# MIRA: Medical Time Series Foundation Model for Real-World Health Data

**Hao Li** [1,2] ♦    **Bowen Deng** [1,3] ♦    **Chang Xu** [1] ♠    **Zhiyuan Feng** [4]
**Viktor Schlegel** [2,6]    **Yu-Hao Huang** [5]    **Yizheng Sun** [2]    **Jingyuan Sun** [2]
**Kailai Yang** [2]    **Yiyao Yu** [4]    **Jiang Bian** [1]

[1] Microsoft Research    [2] University of Manchester    [3] Peking University
[4] Tsinghua University    [5] Nanjing University
[6] Imperial Global Singapore, Imperial College London

## Abstract

A foundation model for medical time series, pretrained on ethically approved clinical datasets, can substantially reduce annotation burdens, minimize the need for task-specific tuning, and promote reliable transferability across healthcare institutions, data modalities, and clinical tasks, especially in data-scarce or privacy-sensitive environments. However, existing generalist time series foundation models struggle to handle medical time series data due to their inherent challenges, including irregular intervals, heterogeneous sampling rates, and frequent missing values. To address these challenges, we introduce **MIRA**, a unified foundation model specifically designed for medical time series forecasting. MIRA incorporates a *Continuous-Time Rotary Positional Encoding* that enables fine-grained modeling of variable time intervals, a *frequency-specific mixture-of-experts layer* that routes computation across latent frequency regimes to further promote temporal specialization, and a *Continuous Dynamics Extrapolation Block* based on Neural ODE that models the continuous trajectory of latent states, enabling accurate forecasting at arbitrary target timestamps. Pretrained on a large-scale and diverse medical corpus comprising over 454 billion time points collect from publicly available datasets, MIRA achieves reductions in forecasting errors by an average of 8% and 6% in out-of-distribution and in-distribution scenarios, respectively, when compared to other zero-shot and fine-tuned baselines. We also introduce a comprehensive benchmark spanning multiple downstream clinical tasks, establishing a foundation for future research in medical time series modeling. Our code is available at `Microsoft/MIRA`.

## 1   Introduction

Medical time series data, including signals such as electrocardiograms (ECG) [1], electroencephalograms (EEG) [2], vital signs and laboratory measurements [3], are key to understanding the dynamic physiological states of patients over time [4, 5]. Continuously modeling these patient data trajectories supports clinical forecasting tasks such as predicting organ failure or treatment response [1, 6], enabling earlier decisions and better patient outcomes [7, 8]. However, effectively utilizing medical time series data in practice remains challenging, as patient populations, disease profiles, and clinical protocols can vary widely across regions and institutions [9]. Moreover, such data is often collected in an irregular and asynchronous manner [10, 11], further complicating efforts to build generalizable

---

♦ Work done during research internship at Microsoft Research.
♠ Corresponding author: `chanx@microsoft.com`.

39th Conference on Neural Information Processing Systems (NeurIPS 2025).

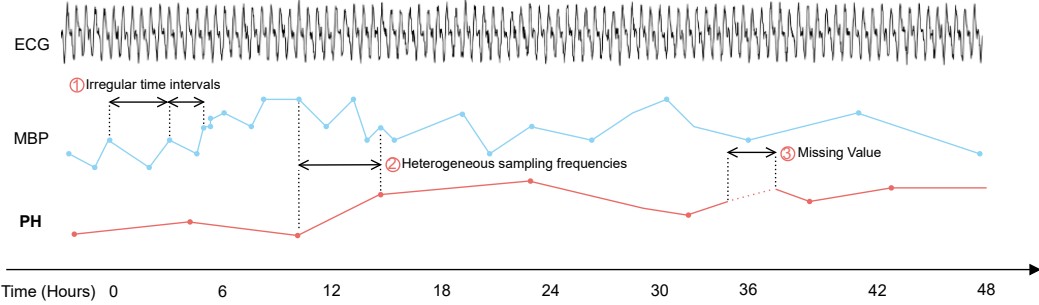

Figure 1: Medical time series exhibit ① irregular intervals, ② heterogeneous sampling rates, and ③ frequent missingness driven by clinical workflows.

models. These challenges are amplified by regulatory frameworks such as GDPR [12, 13], which restrict cross-institutional data exchange and hinder the development of unified modeling paradigms, often resulting in redundant and resource-intensive local optimization efforts. This highlights the need for a foundation model trained on validated medical time series datasets, that is able to learn generalizable patterns from large and varied datasets, reducing the need for extensive data annotation and custom model building, while also helping to share knowledge effectively across different clinical institutions, data types, and medical tasks [14, 15].

However, creating effective medical foundation models for time series is challenging, because clinical data are inherently highly irregular and varied: signals like ECGs recorded at millisecond intervals [16] are found alongside laboratory tests conducted hours apart [17] (Figure 1). Differences in clinical workflows and equipment limits result in irregular sampling, missing values, and inconsistent temporal dependencies [18, 19]. Furthermore, physiological variables differ widely in their value and frequency ranges, requiring models that can handle these varied scales and types of data. Traditional methods using medical task-specific time-series models [20–25], trained on isolated datasets, often do not generalize well to new situations and are difficult to scale [26, 27, 10]. Meanwhile recent general-purpose time series foundation models [28–33] show promise, but they usually assume uniform time intervals. Similarly, emerging medical time series foundation models demonstrate cross-dataset generalization but only within narrowly defined domains—for example, across EEG datasets for sleep monitoring or across datasets for Alzheimer's diagnosis, and cannot handle continuous-time forecasting or irregular sampling [34–37]. These gaps—where current models struggle with the complex, irregular, and multi-frequency nature of medical data [28, 38]—motivates our research and raises a central question:

> *How can we design a scalable and generalizable foundation model that captures the irregularity and multi-frequency nature of medical time series, while enabling robust transfer across diverse medical tasks?*

To address this central question, we introduce **MIRA**—a MEDICAL FOUNDATION MODEL FOR IRREGULARITY ADAPTATION. MIRA is specifically designed to address the challenges of medical time series: First, we propose a *Continuous-Time Rotary Positional Encoding (CT-RoPE)*, which extends the standard RoPE [39] to handle time values using learnable frequency adjustments, allowing MIRA to accurately model the varying frequency intervals of irregularly sampled data. Second, to further improve its ability to adapt to different time features in medical signals, MIRA uses a frequency-specific mixture-of-experts (MoE) layer that learns to route frequency-related information through specialized components. Finally, MIRA integrates a Continuous Dynamics Extrapolation Block, using Neural Ordinary Differential Equations (Neural ODEs) [40]. This feature allows the model to predict patient health paths at any future time point, moving beyond the limits of fixed observation grids. MIRA is pretrained on a large curated collection of medical time series, comprised of over 454 billion time points from publicly available datasets. These datasets cover readings from intensive care units (ICUs), operating rooms, pediatric critical care, and long-term sleep and mental health monitoring. All data undergo preprocessing that standardizes time alignment, normalizes sampling frequencies, and maintains clinical realism. MIRA outperforms other time series foundation models with a similar number of activated parameters across various real-world

benchmarks, achieving reductions in forecasting errors by an average of 8% and 6% in out-of-distribution and in-distribution scenarios, respectively.

To summarize, this paper makes the following contributions: **First**, we propose **MIRA**, a new foundation model specially designed for medical time series forecasting. MIRA directly tackles the challenges of irregular sampling and varied temporal dynamics within a single, well-structured system. **Second**, we curate, preprocess and release an extensive and diverse corpus of medical time series, containing over 454 billion observations, to be used for model pre-training. This collection, built from multiple public datasets, is curated to cover diverse types of medical time series. We select data sources and control their proportions to reflect real-world variability, ensuring correct time alignment, consistent types, and high quality. **Third**, we establish a comprehensive benchmark suite that covers a wide range of clinical forecasting tasks. This suite allows for consistent evaluation of MIRA and future models, encouraging further research and development into robust and generalizable medical time series models.

## 2   Related Work

**Medical Time Series Models.** Machine learning has enabled significant progress in healthcare time series analysis, supporting a wide range of tasks, including dynamic forecasting [21, 22, 41, 42], survival analysis [43, 44], clustering and phenotyping [45–47], screening and monitoring [23, 48, 49], early diagnosis [24, 25, 50], pharmacology [51], treatment effect estimation [52, 53, 42], epidemiological and pandemic influenza surveillance [54], and hospital resource allocation [20, 55]. Despite this progress, most existing methods are narrowly designed for specific tasks or datasets, limiting their adaptability to diverse clinical scenarios [56]. Recently, foundation models for medical time series have emerged, showing promising cross-dataset generalization [34, 37, 35]. However, these models focus primarily on classification tasks and are not designed to handle continuous-time forecasting or irregular temporal patterns common in real-world clinical data.

**Generalist Time Series Foundation Models.** Recent foundation models have shown strong zero-shot forecasting without task-specific tuning [57–61]. Chronos [33] first adapted T5 [62] by discretizing time series intervals as text tokens. More recent models—Moirai [28], Sundial [32], and TimesFM [30]—improve support for variable dimensions and frequencies. Time-MoE [29] and Moirai-MoE [31] further improve specialization with sparse experts. However, these models largely require regularly sampled data, limiting their use on irregular clinical data. See Appendix A for more information.

**Irregular Time Series Models.** Modeling irregular time series has attracted increasing attention, particularly for applications requiring continuous-time reasoning. Early works such as Neural ODEs [40, 63] and State Space Models (SSMs)[64, 65] have demonstrated the ability to capture continuous dynamics by parameterizing the evolution of latent states over time. Another line of research focuses on adapting deep neural architectures, such as RNNs [66–68] and Transformers [19, 69–71], to irregular settings. While these models have shown success in tasks like interpolation and classification, their effectiveness for long-term forecasting remains underexplored.

## 3   Methodology

In this section, we discuss the architecture of MIRA. We first formalize the irregular medical time series forecasting task (Section 3.1). We then introduce the model architecture (Section 3.2). Finally, we present the pretraining corpus and training strategy (Section 3.3).

### 3.1   Irregular Medical Time Series Forecasting

Let each time series instance be represented as paired sequences of timestamps and observations: $\{(t_i, x_i)\}_{i=1}^N$ where $t_i \in \mathbb{R}^+$ are strictly increasing ($t_1 < \cdots < t_N$) and $x_i \in \mathbb{R} \cup \{\text{NaN}\}$ contains real values or missing entries. Our model is designed to handle two prevalent irregular patterns: **(1) Regular-grid missing values.** Observations occur at *equidistant timestamps* $t_i = i\Delta t$ but with *time-level missing values*: some timestamps may have missing measurements. Formally, there exists a subset $\mathcal{M} \subseteq 1, \ldots, N$ such that the corresponding observations $x_i$ are missing (i.e., $x_i = \text{NaN}$) for all $i \in \mathcal{M}$, while the timestamps $t_i$ remain fully observed. **(2) Irregular sampling.** Timestamps

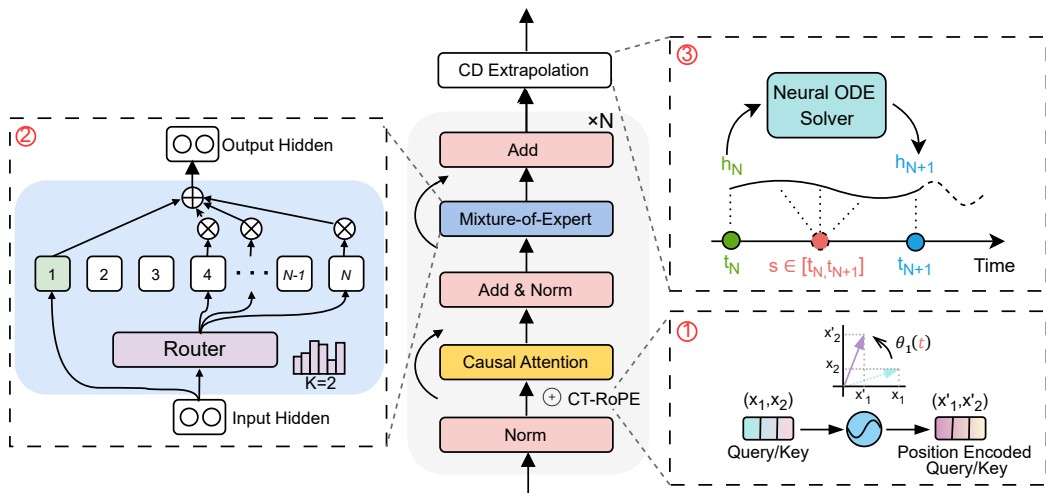

Figure 2: Architecture of **MIRA**. ① Takes irregular medical time series and timestamps as input, applying **CT-RoPE** for continuous temporal encoding. ② A **Sparse Temporal Mixture-of-Experts** layer routes tokens to specialized experts based on frequency. ③ A **Continuous Dynamics Extrapolation Block** evolves latent states toward arbitrary target timestamps for flexible time-aware forecasting.

$t_i$ follow *non-uniform intervals* with $\Delta t_i = t_{i+1} - t_i$ varying for different $i$. All observations at existing timestamps are present ($x_i \in \mathbb{R}$), but the temporal spacing is irregular. Adhering to the channel-independent setting [60], we formulate the problem in a uni-variate time series manner [28]. Given history $\{(x_i, t_i)\}_{i=1}^{L}$, the forecasting task requires predicting future values at **known target timestamps** $\{t_{L+1}, ..., t_{L+H}\}$:

$$\hat{\mathbf{x}}_{L+1:L+H} = f_\theta\big(\{x_i, t_i\}_{i=1}^{L}; \{t_j\}_{j=L+1}^{L+H}\big) \in \mathbb{R}^H \tag{1}$$

where $\{t_j\}_{j=L+1}^{L+H}$ may follow either regularity pattern.

## 3.2 Model Overview

We present **MIRA**, a decoder-only architecture for universal irregular medical time series forecasting. As illustrated in Figure 2, MIRA consists of three key components: (1) a Continuous-Time Rotary Positional Encoding, (2) a Sparse Temporal Mixture-of-Experts Module, and (3) a Continuous Dynamics Extrapolation Block. Together, they enable scalable, frequency-adaptive modeling of non-uniform clinical sequences.

### 3.2.1 Continuous-Time Rotary Positional Encoding

Standard RoPE [39] assumes discrete and uniformly spaced token indices, which limits its applicability to clinical time series with real-valued and irregular timestamps. To address this, we propose a *Continuous-Time Rotary Positional Encoding (CT-RoPE)* that generalizes RoPE to operate directly on continuous-time inputs. Let $t \geq 0$ denote the timestamp of a token in an irregularly sampled sequence, and let $d$ be the model dimensionality, assumed to be even. Specifically, we discretize continuous timestamps $t \geq 0$ into rotation angles without assuming fixed intervals, enabling the model to handle irregular sampling (more detail can be found at Appendix B). The angular frequencies $\{\omega_i\}_{i=0}^{d/2-1}$ are defined as: $\omega_i = 10000^{-2i/d}$. The resulting time-dependent rotation angle is given by:

$$\theta_i(t) = \omega_i \cdot t \tag{2}$$

Given an input embedding $\mathbf{x} \in \mathbb{R}^d$, we partition it into $d/2$ two-dimensional sub-vectors $(x_{2i}, x_{2i+1})$, and apply a planar rotation using the computed angle $\theta_i(t)$:

$$\begin{bmatrix} x'_{2i} \\ x'_{2i+1} \end{bmatrix} = \begin{bmatrix} \cos\theta_i(t) & -\sin\theta_i(t) \\ \sin\theta_i(t) & \cos\theta_i(t) \end{bmatrix} \begin{bmatrix} x_{2i} \\ x_{2i+1} \end{bmatrix} \tag{3}$$

**CT-RoPE with Attention.** The design of the rotary position encoding naturally supports relative position modeling. Specifically, the inner product between rotated query and key vectors depends only on the difference between their timestamps. This can be seen by expanding the attention score:

$$\langle q_m, k_n \rangle = x_m^\top (W^Q)^\top \left( R_\Theta(t_m) \right)^\top R_\Theta(t_n) W^K x_n. \tag{4}$$

Since each $R_{\Theta,i}(t)$ is rotation matrices constructed from trigonometric functions of the timestamp-derived angles, their product is itself a rotation matrix dependent only on the time difference:

$$\left( R_\Theta(t_m) \right)^\top R_\Theta(t_n) = R_\Theta(t_n - t_m), \tag{5}$$

This implies that the positional interaction between two tokens is solely a function of their relative timestamp offset $(n - m)$. In CT-RoPE, this property generalizes to real-valued timestamps, enabling the attention mechanism to capture continuous-time relational structure while retaining the efficiency and structure of standard dot-product attention. In line with Chowdhery et al. [72], we remove biases except in QKV projections to enhance extrapolation. More detail can be found in Appendix C.

### 3.2.2 Frequency-Specific Mixture-of-Experts Block

Medical time series often exhibit dynamics across multiple temporal frequencies, ranging from smooth long-term trends to rapid, short-term variations. To effectively capture such heterogeneity while maintaining computational efficiency, we adopt a sparse Mixture-of-Experts (MoE) architecture to replace the standard feedforward sub-layer in each Transformer block. In practice, each token is routed to a subset of $K$ experts selected from a shared pool of $N$ lightweight feedforward networks. These experts are parameterized independently and are intended to specialize in distinct temporal or semantic structures. Additionally, a *shared expert* is universally applied to all tokens, serving as a global residual pathway. Given the token representation $\bar{\mathbf{u}}_t^l \in \mathbb{R}^D$ at layer $l$, the output of the MoE block is calculated as:

$$\text{MoE}\left(\bar{\mathbf{u}}_t^l\right) = g_{N+1,t} \cdot \text{FFN}_{N+1}\left(\bar{\mathbf{u}}_t^l\right) + \sum_{i=1}^N g_{i,t} \cdot \text{FFN}_i\left(\bar{\mathbf{u}}_t^l\right), \tag{6}$$

where $\text{FFN}_i(\cdot)$ denotes the $i$-th expert network, $\text{FFN}_{N+1}(\cdot)$ the shared expert, and $g_{i,t}$, $g_{N+1,t}$ the corresponding routing weights. The non-shared expert weights $g_{i,t}$ are obtained via a softmax gating mechanism followed by top-$K$ selection:

$$s_{i,t} = \frac{\exp\left((\mathbf{W}_i^l)^\top \bar{\mathbf{u}}_t^l\right)}{\sum_{j=1}^N \exp\left((\mathbf{W}_j^l)^\top \bar{\mathbf{u}}_t^l\right)}, \quad g_{i,t} = \begin{cases} s_{i,t}, & \text{if } s_{i,t} \in \text{TopK}(\{s_{j,t}\}_{j=1}^N, K), \\ 0, & \text{otherwise}, \end{cases} \tag{7}$$

where $\mathbf{W}_i^l \in \mathbb{R}^D$ are the trainable gating vectors. The shared expert weight $g_{N+1,t}$ is computed independently using a sigmoid gate:

$$g_{N+1,t} = \sigma\left((\mathbf{W}_{N+1}^l)^\top \bar{\mathbf{u}}_t^l\right), \tag{8}$$

where $\sigma(\cdot)$ denotes the element-wise sigmoid function and $\mathbf{W}_{N+1}^l \in \mathbb{R}^D$ is the shared expert gating vector.

### 3.2.3 Continuous Dynamics Extrapolation Block

Auto-regressive transformer architectures, which generate predictions in a stepwise manner under causal masking, cannot incorporate the timestamp of the target token during inference, as it is not available until after generation. To address this limitation and enable extrapolation to arbitrary timestamps, we introduce a Neural ODE-based [40] extrapolation module that evolves the latent state from the current token's timestamp to the target prediction token timestamp, allowing time-aware forecasting at unseen or irregular time points.

Given $h(t_N) \in \mathbb{R}^d$ (the state at time $t_N$) and the next target timestamp $t_{N+1}$, the Neural ODE module extrapolates $h_N$ to $h(t_{N+1})$. We define the temporal evolution of the hidden state $h(s)$ over the interval $s \in [t_N, t_{N+1}]$ as:

$$\frac{dh(s)}{ds} = f(s - t_N, h(s); \theta_{ODE}), \quad h(t_N) = h_N, \quad \text{for } s \in [t_N, t_{N+1}] \tag{9}$$

where $f : \mathbb{R}_{\geq 0} \times \mathbb{R}^{D_{model}} \to \mathbb{R}^{D_{model}}$ is the dynamics function, parameterized by a neural network (e.g., an MLP) with parameters $\theta_{ODE}$. $f$ takes the relative time $\Delta s = s - t_N$ and the current state $h(s)$ as input. The state at the target time $t_{N+1}$ is obtained by integrating the ODE dynamics:

$$h(t_{N+1}) = h(t_N) + \int_{t_N}^{t_{N+1}} f(s - t_N, h(s); \theta_{ODE}) ds \tag{10}$$

This integral is computed numerically using an ODE solver (i.e. the Dormand-Prince (RK45) method). Let the result of this numerical integration be denoted as $h'_{N+1} = h(t_{N+1})$. More detail is in Appendix D.

**Implementation with Adaptive ODE Solvers** We numerically approximate the solution to Equation 10 using adaptive step-size ODE solvers. Appropriate absolute and relative error tolerances (e.g., $10^{-6}$) are set to manage the trade-off between accuracy and computational cost.

## 3.3 Model Training

**Medical Pretraining Dataset.** To support generalizable and clinically relevant time series modeling, we curate a large pretraining corpus spanning over 454 billion time points from various real-world healthcare settings. The dataset collection includes signals from ICUs a nd operating rooms, pediatric critical care, long-term sleep and mental health monitoring, and population-level epidemiological surveillance. All data are drawn from publicly available clinical datasets, including MIMIC-III [73], MIMIC-IV [74], PTB-XL [75], Sleep-EDF [76], and the WAVES Pediatric Waveform Database [77]. To enable the model to acquire general knowledge, we resumed training from the Time-MoE checkpoint. A full summary of included datasets and the pre-processing applied is provided in Appendix E.

**Loss Function.** Training large-scale medical time series models requires balancing predictive accuracy, numerical stability, and sparse expert utilization. To ensure robustness against outliers and noisy measurements common in clinical data, we employ the Huber loss $\mathcal{L}_{\text{Huber}}$ over autoregressive multi-horizon predictions:

$$\mathcal{L}_{\text{Huber}}(x, \hat{x}) = \begin{cases} \frac{1}{2}(x - \hat{x})^2, & \text{if } |x - \hat{x}| \leq \delta, \\ \delta \cdot (|x - \hat{x}| - \frac{1}{2}\delta), & \text{otherwise,} \end{cases} \tag{11}$$

where $\delta$ is a threshold controlling the transition between L2 and L1 loss regimes. To avoid expert collapse in the sparse MoE layer, we introduce a load balancing loss $\mathcal{L}_{\text{aux}}$ that promotes uniform usage of experts. Let $f_i$ be the fraction of tokens assigned to expert $i$, and $r_i$ the average routing probability:

$$\mathcal{L}_{\text{aux}} = N \cdot \sum i = 1^N f_i r_i, \tag{12}$$

where $f_i = \frac{1}{KT} \sum_{t=1}^{T} \mathbb{I}(\text{Expert} i \text{ selected at } t)$ and $r_i = \frac{1}{T} \sum t = 1^T s_{i,t}$ with $s_{i,t}$ being the softmax routing score.

Table 1: A high-level summary of MIRA model configurations.

| | Layers | Experts | $K$ | $d_{\text{model}}$ | $d_{\text{ff}}$ | $d_{\text{expert}}$ | Activated Params | Total Params |
|---|---|---|---|---|---|---|---|---|
| $MIRA_{small}$ | 8 | 8 | 2 | 288 | 1152 | 144 | 30 M | 73 M |
| $MIRA_{base}$ | 12 | 8 | 2 | 384 | 1536 | 192 | 50 M | 114 M |
| $MIRA_{large}$ | 12 | 8 | 2 | 768 | 3072 | 384 | 200 M | 455 M |

**Model Configurations.** We adopt a similar model configuration strategy following Time-MoE [29] by providing three model variants of increasing scale: $MIRA_{small}$, with approximately 73 million parameters; $MIRA_{base}$, with approximately 114 million parameters; $MIRA_{large}$, with approximately 455 million parameters;All models are trained on max to eight NVIDIA 80G H/A100 GPUs, using a micro batch size of 128 , and a maximum sequence length of 512. For model configurations and training details, refer to Appendix F.

## 4 Experiments

In this section, we present the empirical evaluation of MIRA. We begin by outlining the experimental setup (Section 4.1). We then evaluate MIRA on zero-shot forecasting benchmarks under both out-of-distribution (Section 4.2) and in-distribution (Section 4.3) settings. Additionally, we analyze the model's scaling behavior across varying model and dataset sizes, and evaluate its robustness to different levels of data irregularity (Section 4.4). Finally, we conduct ablation studies to examine what contributes to MIRA's performance (Section 4.5).

### 4.1 Experiment Setup

**Downstream Datasets.** We evaluate MIRA on a diverse suite of real-world clinical and public health datasets, spanning multi-modal physiological signal analysis, critical care monitoring, ambulatory bio-signal tracking, epidemiological surveillance, and healthcare resource utilization. To systematically study temporal robustness, we group datasets into two categories: (1) *inherently irregular* datasets, which exhibit irregular sampling and natural missing values due to clinical or observational workflows. This are CinC 2012 [78]. Furthermore, we use (2) *originally regular* datasets, i.e.,MIT-BIH [79], Johns Hopkins COVID-19 Dataset [80] ,CDC Influenza Hospitalizations Admissions (CDC-IHA) [1], Heart Rate [81] and illness [82], for which we simulate irregularity by randomly masking 30% of time points. The full list of datasets and statistics is provided in Appendix G.

**Baselines.** We compare MIRA against 13 state-of-the-art forecasting models, which we categorize into three groups: *(i)* zero-shot foundation models, including Time-MoE [29] [2], Moirai [28], Moirai-MoE [31], Moment [83], TimeGPT [84], Timer [85], Lag-Llama [86], TimesFM [30], and Chronos [33], which require inputs to be interpolated for evaluation; and *(ii)* full-shot forecasting models, including ContiFormer [19], T-PatchGNN [87], Neural-CDE [88], and ODE-RNN [66], which are specialized for irregularly sampled time series and require task-specific training. *(iii)* Continue pre-trained zero-shot foundation models on medical corpora, including Time-MoE [29], Moirai [28] and Chronos [33] which performance best on zero-shot evaluation. Implementation details for all baselines are provided in Appendix H.

**Evaluation Metrics.** We measure the Root Mean Squared Error (RMSE) and the Mean Absolute Error (MAE) as the evaluation metrics. Detailed definitions are provided in Appendix I.

### 4.2 Performance on Out-of-distribution Forecasting

**Objective.** We evaluated seven unseen benchmarks excluded from pre-training corpora. For comparison, we fine-tuned the *full-shot forecasting models* on the training split of each benchmark, while the *zero-shot foundation models* were evaluated directly without any task-specific training or fine-tuning. To validate the effectiveness of our model architecture, we additionally pre-trained existing foundation models on the same medical corpus. Notably, *CINC 2012* was excluded from this evaluation for foundation models, as applying regular-grid interpolation at the finest resolution results in over 98% of time steps being interpolated, leading to poor performance for all baselines.

---

[1] https://www.cdc.gov/flu-forecasting/data-vis/current-week.html
[2] We excluded Time-MoE Ultra and Sundial from baselines as they are not open-sourced at the moment.

Table 2: Zero-shot forecasting performance on out-of-distribution datasets that are regularly sampled but contain missing values. Reported values are averaged across all prediction lengths. Lower RMSE and MAE indicate better predictions. **Red**: the best; Blue: the second best compared to zero-shot baselines; Underline: the best performance compared to full-shot baselines.

| Models | Zero-shot Ours | | | | | | Full-shot Time Series Models | | | | | | | | | |
| --- | --- | --- | --- | --- | --- | --- | --- | --- | --- | --- | --- | --- | --- | --- | --- | --- |
| | MIRA$_{small}$ | | MIRA$_{base}$ | | MIRA$_{large}$ | | Contiformer | | T-PatchGNN | | ODE-RNN | | Neural-CDE | | TimesFM | |
| Metrics | RMSE | MAE | RMSE | MAE | RMSE | MAE | RMSE | MAE | RMSE | MAE | RMSE | MAE | RMSE | MAE | RMSE | MAE |
| Cinc 2012 ($10^1$) | 7.136 | 6.984 | 6.762 | 6.734 | 6.221 | 6.115 | 5.987 | 5.985 | 6.247 | 6.246 | 6.997 | 6.995 | 7.498 | 7.497 | - | - |
| Heart Rate ($10^{-1}$) | 1.795 | 1.511 | 1.723 | 1.431 | 1.392 | 1.310 | 0.774 | 0.633 | 0.627 | 0.497 | 0.945 | 0.683 | 0.671 | 0.587 | 1.753 | 0.832 |
| MIT-BIH | 0.293 | 0.198 | 0.199 | 0.141 | 0.173 | 0.130 | 0.453 | 0.354 | 0.705 | 0.627 | 0.882 | 0.623 | 0.242 | 0.196 | 0.335 | 0.141 |
| CDC-IHA ($10^1$) | 5.976 | 4.684 | 5.729 | 4.502 | 5.534 | 4.401 | 5.211 | 4.103 | 9.522 | 7.974 | 10.068 | 9.052 | 7.892 | 6.766 | 15.633 | 4.408 |
| JH COVID-19 ($10^2$) | 0.407 | 0.355 | 0.504 | 0.349 | 0.478 | 0.336 | 0.323 | 0.297 | 0.350 | 0.291 | 0.424 | 0.331 | 0.545 | 0.503 | 2.329 | 0.322 |
| ILI | 1.294 | 1.077 | 1.218 | 1.024 | 1.154 | 1.041 | 0.391 | 0.224 | 0.195 | 0.143 | 0.410 | 0.264 | 0.423 | 0.314 | 2.034 | 1.333 |
| 1$^{st}$ Count | 0 | 0 | 0 | 0 | 4 | 3 | 0 | 0 | 0 | 0 | 0 | 0 | 0 | 0 | 0 | 0 |

| | Zero-shot Foundation Models Pre-trained on General-domain Corpora | | | | | | | | | | | | | | | |
| --- | --- | --- | --- | --- | --- | --- | --- | --- | --- | --- | --- | --- | --- | --- | --- | --- |
| Baseline | Lag-Llama | | TimeGPT | | Timer | | Moment$_{small}$ | | Moment$_{base}$ | | Moment$_{large}$ | | Moirai-MoE$_{small}$ | | Moirai-MoE$_{base}$ | |
| | RMSE | MAE | RMSE | MAE | RMSE | MAE | RMSE | MAE | RMSE | MAE | RMSE | MAE | RMSE | MAE | RMSE | MAE |
| Heart Rate ($10^{-1}$) | 1.764 | 1.488 | 2.258 | 1.915 | 1.901 | 1.704 | 2.966 | 1.852 | 2.939 | 1.835 | 2.917 | 1.735 | 1.982 | 1.600 | 2.144 | 1.652 |
| MIT-BIH | 0.217 | 0.169 | 0.231 | 0.185 | 0.255 | 0.213 | 0.417 | 0.229 | 0.416 | 0.228 | 0.413 | 0.224 | 0.208 | 0.167 | 0.208 | 0.167 |
| CDC-IHA ($10^1$) | 6.531 | 4.846 | 6.654 | 4.860 | 6.424 | 4.857 | 17.803 | 5.307 | 17.631 | 5.288 | 17.689 | 5.260 | 7.099 | 5.405 | 7.639 | 5.862 |
| JH COVID-19 ($10^2$) | 3.596 | 1.432 | 1.879 | 1.580 | 2.647 | 2.328 | 3.077 | 0.553 | 3.097 | 0.554 | 3.064 | 0.549 | 1.452 | 0.725 | 19.391 | 3.190 |
| ILI | 1.780 | 1.366 | 2.011 | 1.077 | 1.882 | 1.492 | 1.570 | 1.056 | 1.566 | 1.054 | 1.566 | 1.052 | 2.001 | 1.678 | 1.983 | 1.664 |
| 1$^{st}$ Count | 0 | 0 | 0 | 0 | 0 | 0 | 0 | 0 | 0 | 0 | 0 | 0 | 0 | 0 | 0 | 0 |

| | Zero-shot Foundation Models Pre-trained on General-domain Corpora | | | | | | | | | | | | | | | |
| --- | --- | --- | --- | --- | --- | --- | --- | --- | --- | --- | --- | --- | --- | --- | --- | --- |
| Baseline | Moirai$_{small}$ | | Moirai$_{base}$ | | Moirai$_{large}$ | | Time-MoE$_{base}$ | | Time-MoE$_{large}$ | | Chronos$_{small}$ | | Chronos$_{base}$ | | Chronos$_{large}$ | |
| | RMSE | MAE | RMSE | MAE | RMSE | MAE | RMSE | MAE | RMSE | MAE | RMSE | MAE | RMSE | MAE | RMSE | MAE |
| Heart Rate ($10^{-1}$) | 2.359 | 1.965 | 2.156 | 1.767 | 2.098 | 1.644 | 0.850 | 0.650 | 0.833 | 0.639 | 1.357 | 0.587 | 1.189 | 0.489 | 1.218 | 0.506 |
| MIT-BIH | 0.343 | 0.249 | 0.421 | 0.302 | 0.593 | 0.149 | 0.171 | 0.135 | 0.172 | 0.135 | 0.353 | 0.147 | 0.361 | 0.149 | 0.350 | 0.147 |
| CDC-IHA ($10^1$) | 6.835 | 5.271 | 7.328 | 5.526 | 6.788 | 5.302 | 6.311 | 4.748 | 6.312 | 4.715 | 15.502 | 4.421 | 15.825 | 4.438 | 15.986 | 4.517 |
| JH COVID-19 ($10^2$) | 1.917 | 0.695 | 0.991 | 0.474 | 0.614 | 0.402 | 0.596 | 0.402 | 0.512 | 0.371 | 4.826 | 1.031 | 3.835 | 0.551 | 3.478 | 0.521 |
| ILI | 1.995 | 1.671 | 1.871 | 1.561 | 1.808 | 1.499 | 1.288 | 0.951 | 1.366 | 1.015 | 2.054 | 1.400 | 1.940 | 1.308 | 1.870 | 1.252 |
| 1$^{st}$ Count | 0 | 0 | 0 | 0 | 0 | 0 | 0 | 0 | 0 | 0 | 0 | 0 | 0 | 0 | 0 | 0 |

| | Zero-shot Foundation Models Continue Pre-trained on Medical Corpora | | | | | | | | | | | | | | | |
| --- | --- | --- | --- | --- | --- | --- | --- | --- | --- | --- | --- | --- | --- | --- | --- | --- |
| Baseline | Moirai$_{small}$ | | Moirai$_{base}$ | | Moirai$_{large}$ | | Time-MoE$_{base}$ | | Time-MoE$_{large}$ | | Chronos$_{small}$ | | Chronos$_{base}$ | | Chronos$_{large}$ | |
| | RMSE | MAE | RMSE | MAE | RMSE | MAE | RMSE | MAE | RMSE | MAE | RMSE | MAE | RMSE | MAE | RMSE | MAE |
| Heart Rate ($10^{-1}$) | 2.047 | 1.685 | 1.907 | 1.536 | 1.601 | 1.263 | 0.648 | 0.524 | 0.603 | 0.488 | 1.049 | 0.555 | 0.965 | 0.488 | 0.902 | 0.458 |
| MIT-BIH | 0.239 | 0.204 | 0.274 | 0.190 | 0.219 | 0.166 | 0.201 | 0.155 | 0.185 | 0.143 | 0.311 | 0.137 | 0.320 | 0.139 | 0.306 | 0.127 |
| CDC-IHA ($10^1$) | 6.690 | 5.310 | 6.934 | 5.376 | 6.696 | 5.114 | 6.327 | 4.698 | 6.299 | 4.666 | 14.502 | 4.321 | 14.825 | 4.338 | 14.986 | 4.417 |
| JH COVID-19 ($10^2$) | 0.579 | 0.448 | 0.619 | 0.478 | 0.812 | 0.337 | 0.517 | 0.362 | 0.517 | 0.362 | 4.225 | 0.947 | 3.535 | 0.510 | 3.328 | 0.469 |
| ILI | 1.528 | 1.289 | 1.435 | 1.191 | 1.501 | 1.229 | 1.188 | 0.915 | 1.201 | 0.927 | 1.705 | 1.127 | 1.639 | 1.081 | 1.547 | 1.028 |
| 1$^{st}$ Count | 0 | 0 | 0 | 0 | 0 | 0 | 0 | 1 | 1 | 1 | 0 | 0 | 0 | 0 | 0 | 0 |

**Results.** Detailed results of out-of-distribution performance are reported in Table 2. First, **MIRA consistently achieves state-of-the-art performance**, outperforming all general-domain foundation models and specialized time series baselines. Specifically, MIRA$_{large}$ achieves the best results on all datasets, improving RMSE by over 8% on average compared to the strongest baselines, confirming the advantage of scaling and medical pretraining. This advantage is particularly pronounced on clinical benchmarks such as MIT-BIH and CDC-IHA, where MIRA$_{large}$ achieves both the lowest RMSE and MAE. Second, **domain-specific pretrnaining proves essential**. All model variants—continue pretrained on medical corpora—consistently outperform models trained on general time series data. This demonstrates the benefit of leveraging medical-specific temporal structures and distributions during pretraining. Interestingly, even smaller MIRA models surpass larger general-domain models, suggesting that data relevance is more critical than model size alone. Third, **MIRA achieves performance close to or even exceeding fine-tuned full-shot models in several cases**. For instance, $MIRA_{large}$ outperform than full-shot baseline on MIT-BIH and slight lower in CDC-IHA.

## 4.3 Performance on In-distribution Forecasting

**Objective.** We evaluated in-distribution performance by holding out a portion of the pre-training datasets as test sets, ensuring no data leakage. All models were tested in zero-shot settings.

**Results.** As shown in Table 3, MIRA consistently achieves highly competitive zero-shot performance across all five pre-training datasets. Compared to baselines such as Moirai, Time-MoE, and Chronos, MIRA demonstrates lower RMSE and MAE on most datasets, particularly excelling on PTB-XL, MIMIC-III, and MIMIC-IV. This advantage holds across all model scales, indicating stable scalability and generalization. In contrast, baselines show larger variance or degradation on challenging datasets

Table 3: Zero-shot forecasting performance on in-distribution datasets that are regularly sampled but contain missing values. Reported values are averaged across all prediction lengths. Lower RMSE and MAE indicate better predictions. **Red**: the best; **Blue**: the second best.

| | Models | Ours | | | Baseline | | | | | | | |
|---|---|---|---|---|---|---|---|---|---|---|---|---|
| | | $MIRA_{small}$ | $MIRA_{base}$ | $MIRA_{large}$ | $Moirai_{small}$ | $Moirai_{base}$ | $Moirai_{large}$ | $Time\text{-}MoE_{base}$ | $Time\text{-}MoE_{large}$ | $Chronos_{small}$ | $Chronos_{base}$ | $Chronos_{large}$ |
| RMSE | SleepEDF ($10^2$) | 0.215 | 0.195 | 0.189 | 0.301 | 0.668 | 0.304 | 0.228 | 0.244 | 0.411 | 0.414 | 0.413 |
| | PTB-XL | 0.147 | 0.127 | 0.121 | 0.177 | 0.270 | 0.416 | 0.110 | 0.109 | 0.228 | 0.234 | 0.229 |
| | MIMIC-III | 0.126 | 0.107 | 0.102 | 0.163 | 0.256 | 0.172 | 0.105 | 0.103 | 0.153 | 0.154 | 0.151 |
| | MIMIC-IV | 0.111 | 0.091 | 0.081 | 0.259 | 0.300 | 0.319 | 0.084 | 0.082 | 0.309 | 0.317 | 0.319 |
| | WAVES | 0.154 | 0.136 | 0.129 | 0.177 | 0.190 | 0.169 | 0.148 | 0.141 | 0.184 | 0.183 | 0.182 |
| | $1^{st}$ Count | 0 | 0 | 4 | 0 | 0 | 0 | 0 | 1 | 0 | 0 | 0 |
| MAE | SleepEDF ($10^2$) | 0.180 | 0.162 | 0.156 | 0.264 | 0.227 | 0.323 | 0.191 | 0.203 | 0.192 | 0.193 | 0.193 |
| | PTB-XL | 0.110 | 0.095 | 0.091 | 0.125 | 0.098 | 0.099 | 0.063 | 0.066 | 0.100 | 0.104 | 0.103 |
| | MIMIC-III | 0.106 | 0.089 | 0.084 | 0.141 | 0.164 | 0.138 | 0.081 | 0.078 | 0.079 | 0.080 | 0.080 |
| | MIMIC-IV | 0.094 | 0.069 | 0.061 | 0.223 | 0.291 | 0.143 | 0.064 | 0.062 | 0.134 | 0.142 | 0.143 |
| | WAVES | 0.129 | 0.112 | 0.106 | 0.157 | 0.181 | 0.155 | 0.124 | 0.116 | 0.119 | 0.118 | 0.117 |
| | $1^{st}$ Count | 0 | 0 | 3 | 0 | 0 | 0 | 1 | 1 | 0 | 0 | 0 |

with missing values. These results validate the robustness of MIRA's architecture in handling imperfect medical time series data without requiring task-specific adaptation.

## 4.4 Model Scaling and Data Behavior.

Table 4: Zero-shot forecasting performance on different missing rates. Reported values are averaged across all prediction lengths. Lower RMSE and MAE indicate better predictions. **Red**: the best; **Blue**: the second best.

| | Missing Rate | 10%+ | | 20%+ | | 30%+ | | 40%+ | | 50%+ | | 60%+ | | 70%+ | | 80%+ | | 90%+ | | $1^{st}$ Count | |
|---|---|---|---|---|---|---|---|---|---|---|---|---|---|---|---|---|---|---|---|---|---|
| | Metrics ($10^2$) | RMSE | MAE | RMSE | MAE | RMSE | MAE | RMSE | MAE | RMSE | MAE | RMSE | MAE | RMSE | MAE | RMSE | MAE | RMSE | MAE | RMSE | MAE |
| Ours | $MIRA_{small}$ | 3.15 | 2.82 | 3.28 | 2.95 | 3.41 | 3.08 | 3.54 | 3.20 | 3.67 | 3.34 | 3.80 | 3.47 | 3.93 | 3.61 | 4.07 | 3.75 | 4.22 | 3.90 | 0 | 0 |
| | $MIRA_{base}$ | 2.98 | 2.68 | 3.11 | 2.81 | 3.25 | 2.94 | 3.38 | 3.07 | 3.51 | 3.20 | 3.64 | 3.34 | 3.78 | 3.48 | 3.93 | 3.63 | 4.10 | 3.79 | 0 | 1 |
| | $MIRA_{large}$ | 2.85 | 2.56 | 2.95 | 2.65 | 3.08 | 2.78 | 3.21 | 2.90 | 3.35 | 3.04 | 3.49 | 3.18 | 3.63 | 3.31 | 3.99 | 3.86 | 3.97 | 3.64 | 8 | 8 |
| General Setting | $Time\text{-}MoE_{base}$ | 3.05 | 2.78 | 3.18 | 2.92 | 3.32 | 3.05 | 3.45 | 3.18 | 3.58 | 3.31 | 3.71 | 3.43 | 3.85 | 3.57 | 3.99 | 3.71 | 4.15 | 3.86 | 0 | 0 |
| | $Time\text{-}MoE_{large}$ | 3.20 | 2.91 | 3.33 | 3.04 | 3.47 | 3.17 | 3.60 | 3.29 | 3.73 | 3.41 | 3.86 | 3.54 | 3.99 | 3.67 | 4.13 | 3.81 | 4.28 | 3.96 | 0 | 0 |
| | $Moirai_{large}$ | 4.80 | 4.18 | 5.39 | 4.64 | 5.68 | 4.55 | 6.02 | 4.83 | 6.09 | 5.06 | 6.18 | 5.02 | 6.31 | 5.09 | 6.51 | 5.24 | 6.71 | 5.36 | 0 | 0 |
| | $Moirai_{base}$ | 4.84 | 4.15 | 4.93 | 4.10 | 5.20 | 4.23 | 5.64 | 4.53 | 5.78 | 4.61 | 5.80 | 4.60 | 5.95 | 4.97 | 5.95 | 4.73 | 6.06 | 4.87 | 0 | 0 |
| | $Moirai_{small}$ | 4.91 | 4.22 | 5.04 | 4.20 | 6.12 | 4.93 | 6.23 | 4.98 | 6.44 | 5.49 | 6.47 | 5.18 | 6.59 | 5.60 | 6.66 | 5.20 | 6.86 | 5.71 | 0 | 0 |
| | $Chronos_{large}$ | 4.71 | 4.10 | 4.85 | 4.22 | 5.02 | 4.31 | 5.31 | 4.52 | 5.58 | 4.70 | 5.82 | 4.91 | 5.97 | 5.10 | 6.15 | 5.32 | 6.43 | 5.49 | 0 | 0 |
| | $Chronos_{base}$ | 5.02 | 4.40 | 5.21 | 4.51 | 5.43 | 4.68 | 5.62 | 4.81 | 5.85 | 4.93 | 6.02 | 5.12 | 6.21 | 5.31 | 6.40 | 5.49 | 6.67 | 5.62 | 0 | 0 |
| | $Chronos_{small}$ | 5.32 | 4.71 | 5.49 | 4.82 | 5.68 | 4.93 | 5.81 | 5.02 | 6.01 | 5.20 | 6.21 | 5.32 | 6.40 | 5.49 | 6.58 | 5.61 | 6.83 | 5.72 | 0 | 0 |
| Medical Setting | $Time\text{-}MoE_{large}$ | 2.95 | 2.70 | 3.08 | 2.83 | 3.22 | 2.96 | 3.35 | 3.09 | 3.48 | 3.22 | 3.61 | 3.35 | 3.75 | 3.49 | 3.89 | 3.63 | 4.04 | 3.78 | 1 | 0 |
| | $Time\text{-}MoE_{base}$ | 3.05 | 2.78 | 3.18 | 2.91 | 3.32 | 3.04 | 3.45 | 3.17 | 3.58 | 3.30 | 3.71 | 3.43 | 3.85 | 3.57 | 3.99 | 3.71 | 4.15 | 3.86 | 0 | 0 |
| | $Moirai_{large}$ | 4.41 | 3.76 | 4.92 | 4.20 | 5.35 | 4.41 | 5.69 | 4.76 | 5.93 | 4.80 | 6.09 | 4.92 | 6.23 | 4.98 | 6.48 | 5.29 | 6.62 | 5.30 | 0 | 0 |
| | $Moirai_{base}$ | 4.55 | 3.92 | 4.78 | 4.07 | 5.18 | 4.26 | 5.49 | 4.46 | 5.78 | 4.64 | 5.70 | 4.51 | 5.87 | 4.97 | 5.92 | 4.75 | 5.96 | 4.71 | 0 | 0 |
| | $Moirai_{small}$ | 5.02 | 4.21 | 5.36 | 4.40 | 5.55 | 4.98 | 5.97 | 4.92 | 6.25 | 5.34 | 6.43 | 5.11 | 6.44 | 5.35 | 6.57 | 5.41 | 6.58 | 5.30 | 0 | 0 |
| | $Chronos_{large}$ | 4.30 | 3.89 | 4.58 | 4.02 | 4.83 | 4.20 | 5.02 | 4.31 | 5.32 | 4.51 | 5.61 | 4.73 | 5.83 | 4.91 | 6.02 | 5.10 | 6.31 | 5.32 | 0 | 0 |
| | $Chronos_{base}$ | 4.61 | 4.10 | 4.83 | 4.20 | 5.02 | 4.31 | 5.32 | 4.51 | 5.58 | 4.70 | 5.82 | 4.91 | 5.97 | 5.10 | 6.15 | 5.32 | 6.43 | 5.49 | 0 | 0 |
| | $Chronos_{small}$ | 4.83 | 4.31 | 5.02 | 4.43 | 5.32 | 4.51 | 5.58 | 4.70 | 5.82 | 4.91 | 5.97 | 5.10 | 6.15 | 5.32 | 6.43 | 5.49 | 6.71 | 5.61 | 0 | 0 |

**Objective.** We investigate the robustness of the MIRA by comparing performance with baselines under varying missing datarates on the WHO FluNet dataset which is not used in pre-training.

**Data Behavior Result.** Table 4 summarizes the zero-shot forecasting performance under varying missing rates, ranging from 10% to 90%. MIRA consistently outperforms all baselines across all missing rates, with $MIRA_{large}$ achieving the lowest RMSE and MAE in nearly every setting. Notably, $MIRA_{large}$ maintains strong performance even as the missing rate increases, showing minimal performance degradation compared to other models. This demonstrates its robustness to severe information loss. While $Time-MoE_{large}$ performs competitively at lower missing rates, its performance drops more quickly as missingness increases.

## 4.5 Ablation Analysis

Table 5: Ablation studies. (**Left**) Evaluated with different model components. (**Right**) Analysis performance and inference speed across different $top_k$ setups. Lower values indicate better performance.

| Component | RMSE | MAE |
|---|---|---|
| $MIRA_{base}$ | 0.154 | 0.118 |
| w/o CT-RoPE | 0.158 | 0.125 |
| w/o MoE Block | 0.157 | 0.122 |
| w/o CT-Extrapolation Block | 0.162 | 0.128 |

| $Top_k$ Setup | RMSE | MAE | Speed (s/iter) |
|---|---|---|---|
| w/ {$Top_1$} | 0.160 | 0.121 | 0.097 |
| w/ {$Top_2$} | 0.154 | 0.118 | 0.101 |
| w/ {$Top_4$} | 0.154 | 0.117 | 0.112 |
| w/ {$Top_6$} | 0.156 | 0.120 | 0.124 |
| w/ {$Top_8$} | 0.159 | 0.122 | 0.127 |

**Objective.** We further perform an ablation study by removing key components of MIRA, to quantify their contribution, and the impact of the number of experts on performance.

**Component Ablation Results.** As shown in the left-hand side of Table 5, removing CT-RoPE (w/o CT-RoPE) leads to a performance drop, confirming the importance of our continuous-time positional encoding. Similarly, eliminating the Mixture-of-Experts block (w/o MoE Block) slightly degrades performance, showing the value of expert specialization. The largest degradation is observed when disabling the CT-Extrapolation Block, with RMSE and MAE increasing by 5 and 8%, respectively, highlighting its role in improving extrapolation. These results showcase the complementary benefits of all three components in achieving robust performance.

**Expert Activation Analysis.**

We further examine the trade-off between predictive performance and inference efficiency by varying the number of activated experts ($top_k$), as shown in the right-hand side of Table 5. Activating two experts ($top_2$) achieves the best balance, reaching the lowest RMSE of 0.154 with an inference speed of 0.101 s/iter. Increasing to four experts provides no meaningful improvement in performance but increases inference time to 0.112 s/iter. Further increasing to six or eight experts leads to slower inference with diminishing returns. Conversely, using a single expert ($top_1$) sacrifices accuracy (0.160) for only a marginal speed gain. These findings indicate that using two experts offers an optimal trade-off between efficiency and accuracy for scalable deployment.

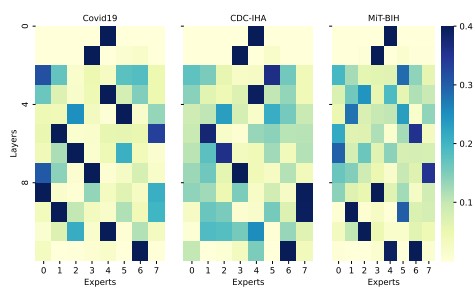

Figure 3: Gating scores for experts across different layers in the three different frequency datasets.

**Activation Visualization.**

We further visualize expert activation patterns on three datasets with distinct temporal resolutions: MIT-BIH (high-frequency, Hz-level), CDC-IHA (weekly), and COVID-19 (daily). As shown in Figure 3, low-frequency datasets such as Covid19 dataset tend to activate a different set of experts compared to the high-frequency MIT-BIH dataset.

## 5   Discussion & Conclusion

We introduce **MIRA**, a foundation model designed for medical time series forecasting under irregular conditions. By integrating CT-RoPE, a Time-Specialized MOE module, and Continuous Dynamics Extrapolation, MIRA demonstrates strong generalization capabilities across diverse medical datasets. This work highlights the potential of scalable and temporally adaptive solutions to real-world medical challenges.

**Limitation.** This work is based on publicly available, de-identified medical datasets. While these resources offer broad coverage and ensure reproducibility, they may not fully reflect the complexities of real-world clinical deployment. In addition, although these datasets are anonymized, residual privacy risks may still exist; however, addressing such risks lies beyond the scope of this work.

## Acknowledgments and Disclosure of Funding

We thank Microsoft Research for providing the computational resources that made this work possible. We are also grateful to the anonymous reviewers for their insightful comments and constructive suggestions, which significantly improved the quality of this paper. Viktor Schlegel is part of the IN-CYPHER programme and is supported by the National Research Foundation, Prime Minister's Office, Singapore, under its Campus for Research Excellence and Technological Enterprise (CRE-ATE) programme. We acknowledge the support from the UMRI IDR Placement 2025 Pioneering project "Foundational AI Models for Brain Computer Interfaces and Neuro-Robotic Control" (PI: Dr. Jingyuan Sun), and The European High Performance Computing Joint Undertaking (EuroHPC JU) with project EHPC-DEV-2025D06-002 (PI: Dr. Jingyuan Sun) and project EHPC-BEN-2025B05-008 (PI: Dr. Jingyuan Sun).

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

# A  Further Related Work

In this section, we delve deeper into the related work on (medical) time series foundation models. Current research efforts in universal forecasting with time series foundation models can be broadly classified into four categories, as summarized in Table 6:

*(i)* **Encoder-only models**. Moirai is trained on the LOTSA dataset comprising 27B time points, with model sizes up to 311M parameters [28]. MOMENT, based on the T5 architecture, is pretrained on the Time-series Pile dataset containing approximately 1B time points, reaching up to 385M parameters [83].

*(ii)* **Encoder-decoder models**, exemplified by Chronos [33], which tokenizes time series data via scaling and quantization, training on both public and synthetic datasets,offers pre-trained models at multiple scales, with the largest containing up to 710M parameters.

*(iii)* **Decoder-only models**. TimesFM is trained on a corpus of 100B time points, with model sizes up to 500M parameters [30]. Lag-Llama focuses on univariate probabilistic forecasting, utilizing a decoder-only Transformer architecture with up to 200M parameters [86]. Timer is a generative pre-trained Transformer model designed for large-scale time series modeling, with a base version containing 84M parameters and pre-trained on 260B time points [85].

*(iv)* **Mixture-of-Experts architectures**. Recent models adopt sparse Mixture-of-Experts (MoE) architectures to enhance scalability and efficiency. Time-MoE [29] scales to 2.4B parameters with only a few experts activated per input, while Moirai-MoE [31] achieves token-level specialization without frequency heuristics, improving adaptability and inference cost.

Table 6: Comparison between time series models.

| Method | MIRA (Ours) | Sundial (2025) | Time-MoE (2024) | Moirai-MoE (2024) | Moirai (2024) | TimesFM (2024) | Moment (2024) | Chronos (2024) | Timer (2024) | Lag-Llama (2023) | TimeGPT (2023) |
|---|---|---|---|---|---|---|---|---|---|---|---|
| Architecture | Decoder | Decoder | Decoder | Decoder | Encoder | Decoder | Encoder | EncDec | Decoder | Decoder | EncDec |
| (Max) Model Size | 455M | 444M | 2.4B | 1.1B | 311M | 500M | 385M | 710M | 67M | 200M | Unknown |
| Input Token | Point | Patch | Point | Patch | Patch | Patch | Patch | Point | Patch | Point | Patch |
| Dataset Scale | 454B | 1TB | 309B | 27B | 27B/231B* | 100B | 1.13B | 84B | 29B | 0.36B | 100B |
| Max Length | 512 | 2880 | 4096 | 5000 | 5000 | 512 | 512 | 512 | 1440 | 1024 | Unknown |
| FFN | Sparse | Dense | Sparse | Sparse | Dense | Dense | Dense | Dense | Dense | Dense | Dense |

* Depend on the way of calculation according to the original paper.

# B  Time Normalization

MIRA operates in continuous time and relies on a rotary positional encoding adapted to irregular timestamps (CT-RoPE). To ensure stable phase computation and consistent temporal scaling across training and autoregressive inference, we apply a deterministic min–max normalization to each full timestamp sequence.

Given a strictly increasing timestamp sequence

$$\mathbf{t} = [t_1, t_2, \ldots, t_T], \tag{13}$$

we compute

$$t_{\min} = \min_i t_i, \qquad t_{\max} = \max_i t_i, \tag{14}$$

and rescale every timestamp into a canonical range that aligns with the sequence length:

$$\widehat{t}_i = \frac{t_i - t_{\min}}{t_{\max} - t_{\min} + \varepsilon}\,(T-1). \tag{15}$$

where, $(T-1)$ corresponds to the maximum index of a sequence of length $T$, ensuring that the normalized time domain matches the index range used in standard rotary encodings, while preserving the original temporal spacing in a continuous form. The small constant $\varepsilon$ prevents numerical instability when the timestamp range is narrow [89, 90].

# C  Mathematical Analysis of CT-RoPE

## C.1  Theoretical Properties

**Linear Angle Scaling.**  For any timestamp $t \geq 0$, the rotation angles grow linearly with time:

$$\theta_i(t) = \omega_i\, t, \tag{16}$$

where $\omega_i = 10000^{-2i/d}$ are fixed angular frequencies.

*Proof.* Directly from Equation (1) in the main text. The linear formulation preserves temporal causality while maintaining bounded rotation magnitudes through exponentially decaying frequencies:

$$\omega_i = \exp\left(-\frac{2i}{d}\ln 10000\right), \tag{17}$$

ensuring higher dimensions receive progressively smaller rotations.

**Relative Position Encoding.**  The inner product between rotated vectors depends only on their temporal difference:

$$\langle R_\Theta(t_i)q,\ R_\Theta(t_j)k\rangle = \langle q,\ R_\Theta(t_j - t_i)k\rangle. \tag{18}$$

*Proof.* Using properties of orthogonal rotation matrices:

$$R_\Theta(t_i)^\top R_\Theta(t_j) = \bigoplus_{k=0}^{d/2-1} \begin{bmatrix} \cos(\omega_k(t_j - t_i)) & \sin(\omega_k(t_j - t_i)) \\ -\sin(\omega_k(t_j - t_i)) & \cos(\omega_k(t_j - t_i)) \end{bmatrix} \tag{19}$$

$$= R_\Theta(t_j - t_i), \tag{20}$$

where we assume $t_j \geq t_i$; otherwise, $R_\Theta(t_j - t_i) = R_\Theta(|t_j - t_i|)$ due to $\cos(-\theta) = \cos(\theta)$ and $\sin(-\theta) = -\sin(\theta)$.

## C.2  Rotation Matrix Construction.

For an even-dimensional model ($d$ even), define the block-diagonal rotation operator:

$$R_\Theta(t) = \bigoplus_{i=0}^{d/2-1} \begin{bmatrix} \cos(\omega_i t) & -\sin(\omega_i t) \\ \sin(\omega_i t) & \cos(\omega_i t) \end{bmatrix}. \tag{21}$$

Key properties include:

- **Norm Preservation:** $\|R_\Theta(t)x\| = \|x\|,\ \forall x \in \mathbb{R}^d$.
- **Temporal Monotonicity:** $t_1 < t_2 \Rightarrow \theta_i(t_1) < \theta_i(t_2)$.
- **Differentiability:** $\frac{\partial R_\Theta(t)}{\partial t}$ exists for all $t > 0$.

## C.3  Attention Mechanism Extension

The scaled dot-product attention becomes:

$$\text{Attention}(Q, K, V) = \text{softmax}\left(\frac{(R_{\Theta, t_Q}Q)(R_{\Theta, t_K}K)^\top}{\sqrt{d}}\right)V \tag{22}$$

where $t_Q$ and $t_K$ are query/key timestamps respectively. This maintains:

- **Temporal Locality**: $\lim_{\Delta t \to 0}\|R_{\Theta, t+\Delta t} - R_{\Theta, t}\| = \mathcal{O}(\Delta t)$
- **Causality**: For $t_Q > t_K$, relative rotations prevent information leakage
- **Efficiency**: Requires only $\mathcal{O}(d)$ extra computation vs standard attention

---
**Algorithm 1** Neural ODE State Transition
---
**Require:** Hidden state $h(t_N)$, timestamps $t_N, t_{N+1}$
**Ensure:** Evolved state $\tilde{h}(t_{N+1})$
 1: Configure ODE solver tolerances: `rtol` $\leftarrow 10^{-6}$, `atol` $\leftarrow 10^{-6}$
 2: Solve ODE: $\tilde{h}(t_{N+1}) \leftarrow$ `odeint`$(f_{\text{ODE}}, h(t_N), [t_N, t_{N+1}])$
 3: **return** $\tilde{h}(t_{N+1})$
---

## D  Neural ODE

**Gradient Computation via Adjoint Sensitivity.** For end-to-end training, gradients of a loss function $\mathcal{L}$ with respect to the ODE parameters $\theta_{\text{ODE}}$ and the initial state $h(t_N)$ are efficiently computed using the adjoint sensitivity method [40]. This involves solving a backward-in-time ODE for the adjoint state $a(s) = \partial\mathcal{L}/\partial h(s)$:

$$\frac{da(s)}{ds} = -\left(\frac{\partial f_{\text{ODE}}}{\partial h}(s, h(s); \theta_{\text{ODE}})\right)^{\top} a(s), \quad a(t_{N+1}) = \frac{\partial\mathcal{L}}{\partial h'(t_{N+1})}. \tag{23}$$

Gradients w.r.t. parameters $\theta_{\text{ODE}}$ are then computed through integrals involving $a(s)$.

**Stability Considerations.** To ensure numerical stability, the dynamics function $f_{\text{ODE}}$ is spectrally normalized, which bounds its Lipschitz constant $L$. This constrains error propagation during integration, yielding a bounded numerical error:

$$\|h_{\text{ODE}}(t_{N+1}) - h_{\text{true}}(t_{N+1})\| \le \frac{M}{L}\left(e^{L\Delta t} - 1\right), \quad \Delta t = t_{N+1} - t_N, \tag{24}$$

where $M$ represents the local truncation error of the ODE solver.

### D.1  Theoretical Guarantees.

Our formulation satisfies the Picard–Lindelöf conditions for existence and uniqueness.

**Theorem 1** (Existence and Boundedness). *Let $f_{ODE}$ be spectrally normalized with maximum singular value $\sigma_{\max}$, and Lipschitz continuous with constant $L = \sigma_{\max}$. Then for $\Delta t = t_{N+1} - t_N > 0$, the state evolution admits a unique solution satisfying*

$$\|\tilde{h}(t_{N+1}) - \tilde{h}(t_N)\| \le \sigma_{\max}\Delta t\, e^{\sigma_{\max}\Delta t}. \tag{25}$$

This bounded evolution property ensures stability for long-horizon extrapolation.

## E  Pretraining Datasets

Table 7: The pre-training dataset of MIRA, which encompasses various sources.

| Source | WAVES (2023) | MIMIC-III (2016) | MIMIC-IV (2020) | Sleep-EDF (2000) | PTB-XL (2020) | Total |
|--------|-------|-----------|----------|-----------|--------|-------|
| # Pts. | 4.8B | 400B | 48B | 0.14B | 1.3B | 454B |
| % | 1.06% | 88.06% | 10.57% | 0.03% | 0.29% | 100.0% |

Our pretraining corpus consists of five publicly available medical time series datasets, selected for their diversity in sampling frequency, modality coverage, and clinical relevance:

**MIMIC-III** [73] is a widely used publicly available database containing de-identified health records of over 40,000 patients admitted to intensive care units (ICUs) at the Beth Israel Deaconess Medical Center between 2001 and 2012. It includes multivariate time series of vital signs, laboratory test results, medication administrations, and clinical notes. The data exhibit irregular sampling patterns and substantial missingness, making them suitable for evaluating temporal robustness.

**MIMIC-IV** [74] is the successor to MIMIC-III, covering ICU and emergency department admissions from 2008 to 2019. It offers expanded variable coverage, improved data standardization, and higher temporal resolution. Time series are recorded with greater granularity across a broader range of hospital units, including both adult and pediatric care. Like MIMIC-III, it reflects realistic clinical irregularity but with enhanced data fidelity and scale.

**PTB-XL** [75] is a 12-lead ECG dataset sampled at 100 Hz, containing over 20,000 clinical recordings annotated with diagnostic statements. It enables robust modeling of waveform-based cardiovascular signals.

**Sleep-EDF** [76] contains overnight sleep EEG and respiration recordings annotated with sleep stages, collected under controlled conditions. The data are sampled at 100 Hz and exhibit moderate regularity with biologically driven transitions.

**WAVES** [77] is a pediatric waveform dataset that combines high-frequency physiological signals (ECG, PPG, respiration) from intensive care and operating room settings. It provides long continuous sequences with natural variability and noise.

## F   Model Configurations and Training Detail

Informed by recent scaling studies [29], we design MIRA in three model scales: $MIRA_{small}$ (73M parameters), $MIRA_{base}$ (114M), and $MIRA_{large}$ (455M), enabling flexible deployment across different hardware constraints. All models are trained on up to eight NVIDIA 80GB A100 GPUs with a micro-batch size of 128 and a maximum sequence length of 512. We pre-trained one epoch with each training step processes approximately 65,000 time points. We consider forecast horizons of $24, 32, 48, 64$ for short-term and long-term evaluation. Following standard practice, we apply an auxiliary load balancing loss with weight $\alpha = 0.02$ to encourage expert utilization.

## G   Downstream Tasks and Datasets

**CinC 2012**[78] originates from the PhysioNet 2012 Challenge and contains multivariate time series from ICU patients. Each record includes asynchronously sampled clinical variables such as blood pressure, heart rate, and oxygen saturation, with irregular measurement frequencies and observation patterns. The forecasting task focuses on predicting future physiological values and deterioration risk based on past observations.

**CDC-IHA**[3], published by the U.S. CDC, contains weekly aggregated hospital admission metrics related to respiratory diseases across U.S. jurisdictions. It includes counts for COVID-19, influenza, and RSV-related hospitalizations. The signals are inherently discrete and asynchronous across regions, with missing entries due to delayed or inconsistent reporting. The forecasting task involves predicting near-future hospitalization counts by location. We using below column in an weekly frequency: *Total Patients Hospitalized with COVID-19, Total Patients Hospitalized with Influenza,Total Patients Hospitalized with RSV, Total ICU Patients Hospitalized with COVID-19,Total ICU Patients Hospitalized with Influenza,Total ICU Patients Hospitalized with RSV.*

**MIT-BIH** [79] This dataset provides annotated ECG waveform segments from patients with arrhythmias. The input is a regularly sampled 2-lead ECG sequence, and the forecasting task involves predicting future signal windows. We introduce synthetic missingness during evaluation to test robustness.

**JHU COVID-19 Dataset** [80] This dataset aggregates daily COVID-19 cases and deaths by region. While data are reported at regular intervals, inconsistencies and reporting delays motivate the use of masked evaluation. Forecasting focuses on short-term case count prediction. We using this dataset in a daily frequency

**CDC-Illness** [82] The CDC outpatient illness dataset tracks ILI (influenza-like illness) and other symptoms across U.S. reporting centers. Although data are uniformly weekly, we mask 30% of time points to simulate partial surveillance dropout.

---

[3]https://data.cdc.gov/Public-Health-Surveillance

**Heart Rate** This dataset, sourced from the Monash Time Series Extrinsic Regression Archive [81], contains photoplethysmography (PPG) sequences paired with continuous heart rate values as targets. Each instance represents a short PPG segment with the goal of predicting the corresponding heart rate, making it a benchmark for physiological signal regression tasks.

## H  Baselines

**Time-MoE** [29]: A sparsely activated decoder-only Transformer model that incorporates a Mixture-of-Experts (MoE) architecture with token-level routing. Trained on over 300 billion real-world time points spanning nine domains, Time-MoE demonstrates strong zero-shot forecasting performance, especially in long-range and multi-resolution prediction. It uses autoregressive decoding and sliding window inference with shared expert regularization.

**Moirai** [28]: A universal forecasting backbone that uses patch-wise tokenization and any-variate self-attention. It is trained on the LOTSA dataset comprising 27 billion time points and supports forecasting across arbitrary time steps, variable sets, and resolutions. Moirai also employs resolution-adaptive projection layers to accommodate different patch sizes during inference.

**Moirai-MoE** [31]: An extension of Moirai that integrates a token-level MoE module into the decoder block. This allows for frequency-specific specialization without relying on handcrafted signal partitions. Moirai-MoE achieves improved generalization across domains with limited added compute, and supports both token-aware routing and auxiliary balancing loss.

**TimesFM** [30]: A decoder-only Transformer developed by Google Research, pre-trained on 100 billion real-world time points from diverse sources including IoT, finance, and weather. It features autoregressive generation with fixed-length context windows and shows strong performance in forecasting across hundreds of benchmarks.

**Chronos** [33]: A family of probabilistic time series foundation models that transform numerical sequences into quantized token representations. Chronos leverages discrete latent spaces and causal language modeling for forecasting and sampling. It supports both point and probabilistic prediction.

**Timer** [85]: A generative time series language model trained on a diverse mix of real-world and synthetic data. It uses next-token prediction and masked time-step modeling for flexible downstream adaptation. While architecture and scale details are less publicly documented, it has demonstrated competitive zero-shot accuracy on common forecasting benchmarks.

**ContiFormer** [19]: A continuous-time Transformer model that combines Neural ODEs with attention. It uses a NeuralODE kernel for modeling value transitions between observations and a standard self-attention block for relational reasoning. It is trained end-to-end on irregularly sampled series.

**T-PatchGNN** [87]: A graph-based model that converts univariate time series into a graph of overlapping time patches, using GNNs to capture local and global dependencies. It is especially effective for sparse and low-signal series and supports continuous-time node embedding.

**Neural-CDE** [88]: A continuous-time model based on controlled differential equations. It encodes the trajectory of observed data using a Neural CDE solver and predicts future values through learned hidden dynamics. Particularly suited for datasets with asynchronous observations.

**ODE-RNN** [66]: Combines standard RNN encoders with ODE solvers to update latent states over irregular intervals. It supports interpolation between time steps and improves temporal continuity compared to discrete RNNs.

**Moment** [83] presents open-source Transformer-based models pre-trained on the Time-series Pile, a large and diverse collection of public time series data. Employing a masked time series prediction task, MOMENT demonstrates strong performance across forecasting, classification, and anomaly detection tasks, particularly in low-resource settings .

**Lag-Llama** [86] is a decoder-only Transformer model tailored for univariate probabilistic forecasting. By incorporating lagged inputs as covariates and pretraining on a diverse corpus of time series data, Lag-Llama exhibits robust zero-shot generalization and state-of-the-art performance upon fine-tuning on unseen datasets .

**TimeGPT** [84] is introduced as the first foundation model for time series analysis, capable of generating accurate predictions across diverse datasets without additional training. Leveraging advancements in deep learning, TimeGPT's zero-shot inference outperforms traditional statistical, machine learning, and deep learning methods in both performance and efficiency .

## I   Evaluation Metrics

We evaluate forecasting performance using two widely adopted metrics: Mean Absolute Error (MAE) and Root Mean Squared Error (RMSE). Both are computed over all predicted time steps and variables, averaged across evaluation windows.

**Mean Absolute Error (MAE)** quantifies the average absolute deviation between predictions $\hat{x}_t$ and ground truth $x_t$:

$$\text{MAE} = \frac{1}{N} \sum_{t=1}^{N} |x_t - \hat{x}_t|, \tag{26}$$

where $N$ denotes the number of valid (unmasked) prediction points. MAE is robust to outliers and provides a direct measure of average deviation.

**Root Mean Squared Error (RMSE)** emphasizes larger errors by squaring deviations before averaging:

$$\text{RMSE} = \sqrt{\frac{1}{N} \sum_{t=1}^{N} (x_t - \hat{x}_t)^2}. \tag{27}$$

It is more sensitive to large prediction errors and penalizes high-variance outputs.

