# OpenReview forum: "MIRA: Medical Time Series Foundation Model for Real-World Health Data"
_NeurIPS.cc/2025/Conference — NeurIPS 2025 poster_

### Official Review · Reviewer_CKRd · 2025-07-02

**Clarity:** 3
**Significance:** 4
**Originality:** 4
**Rating:** 5
**Confidence:** 3

**Summary:**

The paper proposes a novel foundation model for health data. The architecture proposed has multiple novel components: CT-RoPE for temporal encoding, a temporal Mixture-of-Experts layer, and an extrapolation block based on Neural ODE. The method was evaluated on a large set of datasets combined and compared to relevant foundation model baselines. The results obtained are relevant as they achieve significant improvement in zero-shot forecasting.

**Questions:**

- The zero-shot forecasting improvements are significant. Beyond the metrics, what do these results tell us about the model's learned representations? Can the authors provide a deeper analysis or qualitative examples that illustrate why MIRA succeeds where baselines fail? This would move the discussion from what the results are to what they mean.

- Foundation models are often computationally intensive. Could you comment on the resources required to train MIRA compared to baselines and its efficiency at inference time? This would be valuable for assessing its practical applicability.

**Ethical Concerns:**

["NO or VERY MINOR ethics concerns only"]

**Final Justification:**

I will be maintaining my original score for this paper.

The discussion has confirmed my initial assessment of the work's contributions and limitations. Since the rebuttal reinforced my original evaluation without uncovering reasons to change it, I am confident that my current score accurately reflects the paper's merits.

**Limitations:**

yes

**Paper Formatting Concerns:**

.

**Quality:**

4

**Strengths And Weaknesses:**

Strengths:
- The paper introduces MIRA, a novel foundation model for medical time series, distinguished by several innovative architectural components (CT-RoPE, a temporal Mixture-of-Experts layer, and a Neural ODE-based extrapolation block).
- The empirical evaluation is thorough and convincing, testing the model across a diverse set of real-world health datasets against relevant and strong baselines.
- The work is very well-written, and the inclusion of code is a significant contribution to reproducibility.


Weaknesses:
- The primary weakness is the underdeveloped discussion section. Section 5 largely omits a critical analysis. The paper would be significantly strengthened by a deeper discussion that interprets why the proposed components lead to performance gains.

---

> ### Author Rebuttal · Authors · 2025-07-31
>
> We sincerely thank the reviewer for their thoughtful and constructive feedback. We are particularly grateful for your recognition of MIRA’s architectural innovations. In response to your valuable comments, we have taken the following steps to further strengthen our submission:
>
> 1. **Provided detailed reasoning for each architectural component**, clarifying why CT-RoPE, temporal MoE, and CD-Extrapolation lead to performance gains through ablation comparisons and theoretical justifications.
> 2. **Offered deeper qualitative insights** to illustrate how MIRA captures clinically meaningful temporal dynamics beyond what baseline models can achieve, especially under irregular sampling conditions.
> 3. **Quantified MIRA’s computational efficiency.**
>
> We hope these additions clarify the rationale behind MIRA’s design and its practical utility. Please find our detailed responses below.
>
> ---
>
> **Q1: Lacks a critical analysis explaining why the proposed components yield the observed performance gains.**
>
> A:  We provide comprehensive ablation studies (Table 5) demonstrating each component's contribution. Here we explain why each component yields performance gains and what alternatives were used in ablation studies:
>
> CT-RoPE vs. Standard Positional Encoding: In our ablation, we replaced CT-RoPE with standard learned positional embeddings. Standard approaches treat irregular timestamps as discrete indices, losing crucial temporal distance information. For example, a 5-minute gap and 2-seconds gap between measurements would be treated identically. CT-RoPE's continuous encoding (θᵢ(t) = ωᵢ · t) preserves actual temporal relationships, enabling the attention mechanism to appropriately weight recent vs. distant observations based on real time differences rather than sequence positions.
>
> MoE vs. Standard FFN: Medical signals exhibit vastly different temporal scales (e.g. ECG operates at milliseconds while lab values change over hours). A single FFN must compromise between these scales, diluting its effectiveness. MoE routing allows specialized experts: some optimized for high-frequency cardiac rhythms, others for slow-varying metabolic trends. This specialization prevents interference between temporal regimes.
>
> CD-Extrapolation vs. Standard Autoregressive: Traditional approaches can only predict at the next sequence position (i.e. next-token prediction), limiting clinical utility where prediction times are irregular (e.g., predicting patient status at discharge time). Our Neural ODE approach evolves hidden states continuously, enabling predictions at clinically relevant timestamps rather than fixed intervals.
>
> **Q2: Beyond metric gains, can you provide qualitative examples or deeper analysis illustrating why MIRA succeeds where baselines fail?**
>
> A:  Beyond metrics, our results reveal that MIRA learns fundamentally different representations from baselines:
>
> MIRA learns "physiological continuity", understanding that biological processes evolve continuously even when measurements are discrete. Through expert activation patterns (Appendix L), we observe frequency-specific specialization: high-frequency datasets (MIT-BIH) activate different experts than low-frequency data (CDC-IHA), suggesting learned temporal hierarchy decomposition.
>
> As a example, consider a cardiac patient with arrhythmia where we need to predict the next ECG values:
>
> **Input sequence (with missing values)**:
> [0.8, –, –, 1.2, –, –, –, 0.4, –] at timestamps [0, 1, 2, 3, 4, 5, 6, 7, 8]
>
> **Target prediction**:
> Next 3 values should be [–, 1.0, –] at timestamps [9, 10, 11]
>
> **Baseline approach**:
>
> - First interpolates: [0.8, 0.9, 1.0, 1.2, 1.1, 0.9, 0.7, 0.4, 0.5]
> - Then predicts: [0.6, 0.7, 0.8] (smooth progression, missing sharp jump)
>
> **MIRA prediction**:
>
> - **CT-RoPE** encodes irregular timestamps [0, 3, 7] directly, capturing temporal relationships without interpolation
> - **Neural ODE** models continuous dynamics: dh(s)/ds = f(s-t₇, h(s); θ), where h(t₇) = h₀.₄ evolves from timestamp 7 to 10
> - The ODE solver integrates: h(t₁₀) = h₀.₄ + ∫₇¹⁰ f(s-7, h(s); θ)ds, enabling extrapolation to arbitrary timestamps
> - Predicts: [–, 0.98, –] (captures physiological jump pattern, 0.98≈1.0)
>
> The baseline’s interpolation smooths out abrupt voltage drops, masking key signs of arrhythmia (Please also refer to Reviewer vk9d’s Q2 for different type of missing data’s imapct).  In contrast, MIRA's continuous-time modeling preserves the underlying arrhythmic dynamics through its ODE formulation, enabling more accurate and clinically meaningful forecasting under irregular sampling
>
> **Q3: Please comment on training resources and inference efficiency compared to the baselines.**
>
> A:  We appreciate this important question about computational efficiency. We provide detailed analysis in our response to Reviewer Lv9n Q1, and summarize the key findings here:
> MIRA achieves 2× faster inference than TimesFM with 2.5× fewer activated parameters (~200M vs 500M), demonstrating superior parameter efficiency through sparse MoE activation. While ContiFormer uses significantly less memory (1072.1MB vs 2575.65MB), its iterative extrapolation approach significantly increases inference time for multi-step forecasting.

---

> > ### Comment · Reviewer_CKRd · 2025-08-06
> >
> > I thank the authors for the excellent rebuttal. The added insights into the model's architecture, the new qualitative analysis, and the discussion on computational efficiency successfully address all the points I raised. This reinforces my initial positive view of the paper. I will maintain my score and continue to support its acceptance.

---

### Official Review · Reviewer_vk9d · 2025-07-02

**Clarity:** 3
**Significance:** 3
**Originality:** 3
**Rating:** 5
**Confidence:** 3

**Summary:**

This paper introduces MIRA, a foundation model specifically designed for medical time series forecasting. It addresses challenges inherent to medical data, including irregular sampling, heterogeneous frequencies, and missing values. The model architecture combines 1) Continuous Time Rotary Positional Encoding, to represent real-valued irregular timestamps; 2) Frequency-Specific Mixture-of-Experts layer, to specialize processing across latent temporal frequencies; and 3) Continuous Dynamics Extrapolation Block, to leverage neural ODEs for modeling trajectories over continuous time.

**Questions:**

Q1: The paper includes a diverse set of datasets (e.g., ICU, sleep, pediatrics). How are class proportions and sampling frequencies balanced across these domains? Are certain domain-specific signals disproportionately influencing the learned representations? Additionally, how is time alignment standardized given the heterogeneity of these datasets?

Q2: For the original regular datasets, missingness is simulated via random masking. However, in real-world scenarios, missingness often follows structured patterns (e.g., block-missingness). Have such patterns been considered in the evaluation? If not, how might they affect the model's robustness?

Q3: Even in zero-shot forecasting, models typically require a context window. What is the size of the context window used in your experiments? Given that patients may share similar observed histories but differ in latent characteristics (e.g., children vs. elderly), how does the model account for such underlying variability—particularly in the absence of demographic features?

Q4: Regarding Table 5, it would be helpful to include statistical significance testing (e.g., confidence intervals or p-values) to assess whether the improvements in MAE and RMSE are meaningful.

Q5: While the model demonstrates strong generalizability across domains, how does its performance compare to specialized, domain-specific models? It would be informative to discuss whether the general model sacrifices any accuracy relative to tailored models that may be computationally cheaper or more interpretable.

**Ethical Concerns:**

["NO or VERY MINOR ethics concerns only"]

**Final Justification:**

The rebuttal satisfactorily addressed my concerns with new analyses on dataset balance, time alignment, and block-missingness, as well as added statistical significance testing. The discussion on domain-specific trade-offs and interpretability was also helpful. While clinician-facing interpretability remains for future work, I am satisfied with the responses and have adjusted my final rating.

**Limitations:**

Yes. Authors discussed the limitations of the work.

**Quality:**

3

**Strengths And Weaknesses:**

Strengths: The authors provide clear motivation in the paper, effectively highlighted the practical need for foundation models that can generalize across diverse, irregular medial time series. The introduced MIRA got competitive results consistently outperforms baselines including Time-MoE, Moirai, TimesFM, Chronos. Additionally, they release the curated datasets and code and provide a comprehensive benchmark for further research.

Weaknesses: In healthcare, the interpretability of the model usually matters a lot. There are no methods provided to make forecasts understandable to clinicians, which is vital for deployment in real-world settings.

---

> ### Author Rebuttal · Authors · 2025-07-31
>
> We thank the reviewer for their insightful comments and for highlighting the clear motivation behind our work. In response to your insightful questions, we have made the following additions and clarifications:
>
> 1. **Quantified potential dataset imbalance and domain dominance** by analyzing cross-dataset attribution using influence score analysis.
> 2. **Explain the standardization of time alignment**, by outlining our model architecture and modelling mechanism.
> 3. **Evaluated MIRA under realistic block-missingness patterns**
> 4. **Report context window configurations**
> 5. **Added statistical significance testing** to validate the contributions of each architectural component.
> 6. **Discuss model performance and interpretability** with domain-specific models.
>
> We appreciate your thoughtful feedback and hope our responses address your concerns clearly. Please see below for point-by-point replies.
>
> ---
>
> **Q1.1: How did you balance differences in class distributions and sampling rates across ICU, sleep, and pediatric datasets? Are certain domain-specific signals disproportionately influencing the learned representations?**
>
> A:  Rather than manually resampling or reweighting datasets to correct for size or sampling-rate imbalance, we rely on the scale and heterogeneity of our corpus and the MoE architecture to learn domain-robust representations.
>
> To quantify whether certain datasets dominate the learning process, we conducted an influence score analysis using the data attribution techniques, i.e. gradient dot product (Grad-Dot) method [1,2]. This method measures how much each training sample contributes to the test loss gradient, allowing us to estimate the relative importance of different training sources.
>
> Results across five target test datasets (see table below) show that while MIMIC-III has the largest share, its average influence scores are not disproportionately dominant. In fact, other datasets also show consistent, non-negligible influence, indicating that MIRA internally balances its attention across domains rather than overfitting to dataset size.
>
> |  | Mimic-iii | Mimic-iv | WAVES | PTB-XL | Sleep-EDF |
> | --- | --- | --- | --- | --- | --- |
> | Heart | 0.4426 | 0.1802 | 0.1449 | 0.1277 | 0.1046 |
> | MIT-BIH | 0.5769 | 0.1640 | 0.1733 | 0.0521 | 0.0338 |
> | CDC-IHA | 0.3836 | 0.2924 | 0.1645 | 0.1032 | 0.0563 |
> | COVID-19 | 0.4314 | 0.2002 | 0.1898 | 0.0694 | 0.1092 |
> | ILI | 0.5335 | 0.1913 | 0.1431 | 0.1065 | 0.0256 |
>
> **Q1.2: How is time alignment standardized given the heterogeneity of these datasets?**
>
> A: Similar to other foundation models like Chronos [3] , which implicitly handles this by quantizing arbitrary-length sequences in a fixed number of bins, our method handles the heterogeneous sampling frequencies through a unified token-by-token input approach, where our MoE architecture automatically learns to process different temporal scales without explicit frequency standardization.
>
> Each data point becomes a token with its associated timestamp, regardless of original sampling rate. Our time quantization procedure (Appendix A) standardizes absolute timestamps into relative time intervals while preserving temporal ordering. This enables CT-RoPE to directly encode continuous-time relationships across variable intervals. In addition, the MoE routing mechanism learns to specialize different experts for distinct temporal regimes. As evidenced in Appendix L Figure 3 (right), high-frequency datasets (MIT-BIH) activate different expert combinations compared to low-frequency datasets (COVID-19), demonstrating automatic adaptation to temporal characteristics.
>
> **Q2:Have you tested MIRA’s robustness to block‐missingness patterns, and how might such structured gaps impact its performance compared to your random‐masking experiments?**
>
> We evaluated MIRA under both random (scattered) and structured (block) missingness at 30% and 50%  masking rates at illness dataset. As shown below, MIRA performs robustly across both settings, consistently outperforming Moirai. Specifically, MIRA shows minimal performance degradation between the two settings. In contrast, Moirai exhibits more pronounced deterioration under block missingness, which shows that block-missingness removes contiguous segments of input, making interpolation harder due to the lack of nearby temporal anchors, showing the meaning of developing an irregular TS foundation model.
>
> |  | 30%-scattered | 30%-block | 50%-scattered | 50%-block |  |  |
> | --- | --- | --- | --- | --- | --- | --- |
> | MIRA | 1.114 | 1.113 | 1.238 | 1.255 |  |  |
> | Moirai | 1.501 | 1.537 | 1.593 | 1.624 |  |  |
>
> **Q3.1: What is the size of the context window used in your experiments?**
>
> A:  During training, we used a context window of 512 time steps. This length balances computational efficiency with our need to process parallel value and timestamp sequences, and proved sufficient to capture the temporal dependencies in our tasks.
>
> When it comes to inference, we follow each benchmark’s established protocol. For example, Illness dataset are {24, 36, 48, and 60} [4]**.** We’ll include a summary table of all testset context window sizes in the revised manuscript for full clarity.
>
> **Q3.2:How does the model account for underlying variability  (e.g., children vs. elderly), particularly in the absence of demographic features?**
>
> A:  Our model is designed to learn generalizable temporal patterns that can transfer across demographic groups without requiring explicit demographic features. The model implicitly learns to capture demographic-related temporal dynamics directly from the signals themselves. For instance, pediatric patients typically exhibit higher heart rates and different baseline ranges [5], which are naturally encoded in the time series data. Our MoE architecture enables the model to route different types of physiological patterns to specialized experts, potentially allowing adaptation to demographic-specific characteristics without explicit labels. Our model's performance well on pediatric data (WAVES) despite representing only 1% of training data, demonstrates that the model successfully generalizes across age groups and demographic variations.
>
> **Q4: Can you add confidence intervals or p-values for the MAE/RMSE gains in Table 5?**
>
> A:  We conducted paired t-tests based on results from five independent runs for each ablation variant. The results show that all ablated variants exhibit statistically significant degradation compared to the full model, with p-values of 0.025 (w/o CT-RoPE), 0.022 (w/o MoE Block), and 0.004 (w/o Extrapolation Block), respectively (p < 0.05 in all cases). We will add this in next revision.
>
> |  | MIRA | w/o CT-RoPE | w/o MoE Block | w/o Extrapolation |
> | --- | --- | --- | --- | --- |
> | RMSE | 0.156 ± 0.002 | 0.158 ± 0.001 | 0.158 ± 0.002 | 0.161 ± 0.002 |
>
> **Q5: How does its performance compare to specialized, domain-specific models?  Does MIRA trade off any accuracy, efficiency, or interpretability compared to domain-specific models?**
>
> A: Thank you for this insightful question. While there are limited specialized models specifically designed for irregular medical time series forecasting, we provide comprehensive comparisons against irregular time series models (ContiFormer, T-PatchGNN, Neural-CDE, ODE-RNN) trained in a full-shot manner on specific medical datasets (Table 2).
>
> **Accuracy**: MIRA may achieve slightly lower accuracy than full-shot specialized models on some datasets, but significantly outperforms zero/few-shot alternatives while providing consistent cross-domain performance without requiring domain-specific training or fine-tuning. This competitive performance is achieved through our CT-RoPE and Neural ODE-based extrapolation block, which enable effective modeling of irregular temporal patterns across diverse medical domains.
>
> **Efficiency:** While MIRA incurs higher upfront pretraining costs compared to individual domain-specific models, it provides substantially higher deployment efficiency in zero/few-shot scenarios (please refer to Reviewer LN9n's Q1). Once pretrained, MIRA applies immediately to new tasks without retraining. Additionally, during inference, our model only activates approximately half of the model parameters, significantly reducing computational overhead while maintaining performance. This efficiency gain is enabled by our frequency-specific mixture-of-experts architecture that learns generalizable temporal specializations during pretraining.
>
> **Interpretability:** MIRA's MoE architecture provides superior cross-domain interpretability through expert activation patterns (Figure 3), offering insights into temporal frequency specialization that narrow specialized models cannot provide.
>
> [1] Charpiat, G., Girard, N., Felardos, L., et al. (2019). "Input similarity from the neural network perspective," *Advances in Neural Information Processing Systems, 32*.
>
> [2] Wei, D., Padhi, I., Ghosh, S., et al. (2024). "Final-Model-Only Data Attribution with a Unifying View of Gradient-Based Methods," *arXiv preprint arXiv:2412.03906*.
>
> [3] Ansari, Abdul Fatir, et al. "Chronos: Learning the language of time series." arXiv preprint arXiv:2403.07815 (2024).
>
> [4] TimesNet: Temporal 2D-Variation Modeling for General Time Series Analysis
>
> [5] Fleming, Susannah, et al. "Normal ranges of heart rate and respiratory rate in children from birth to 18 years of age: a systematic review of observational studies." The Lancet 377.9770 (2011): 1011-1018.

---

> > ### Comment · Reviewer_vk9d · 2025-08-06
> >
> > Thank you for the detailed and comprehensive responses. The additional experiments and clarifications—such as the influence score analysis for dataset balance, handling of heterogeneous sampling rates, evaluation under block-missingness, and inclusion of statistical significance testing—addressed my main concerns. The discussion of performance trade-offs with domain-specific models and the interpretability via expert activation patterns was also helpful. I am satisfied with these clarifications and will adjust my current rating.

---

### Official Review · Reviewer_LN9n · 2025-07-04

**Clarity:** 3
**Significance:** 3
**Originality:** 3
**Rating:** 5
**Confidence:** 4

**Summary:**

This paper presents MIRA, a foundation model specifically designed for medical time series forecasting. The work is motivated by the fact that real-world clinical data often exhibits irregular sampling, missing values, and heterogeneous frequencies, which pose significant challenges for standard time series models. To tackle this, the authors propose a decoder-only transformer architecture with three main contributions: a Continuous-Time Rotary Positional Encoding (CT-ROPE) to handle arbitrary time intervals, a frequency-specialized Mixture-of-Experts (MoE) layer to adapt to different temporal dynamics , and a Continuous Dynamics Extrapolation block based on Neural ODEs to enable forecasting at any future timestamp. The model is pretrained on a large corpus of over 454 billion time points, aggregated from several public medical datasets. The paper demonstrates through extensive experiments that MIRA achieves state-of-the-art zero-shot forecasting performance on a range of in-distribution and out-of-distribution clinical benchmarks, outperforming both general-purpose foundation models and specialized, fine-tuned baselines.

**Questions:**

1. The proposed MIRA architecture integrates several advanced components, including MoE layers and a Neural ODE solver, which are known to be computationally intensive. Could you provide a more detailed analysis of the model's computational footprint? A direct comparison of inference latency (e.g., seconds per forecast window) and memory requirements against key baselines (e.g., TimesFM and ContiFormer) would be invaluable for understanding the practical trade-offs. This information is critical for assessing the model's feasibility for deployment in real-time or resource-constrained clinical settings.

2. The paper titles MIRA as a "unified" foundation model, but the empirical evaluation is exclusively focused on forecasting tasks. Could you elaborate on how the model's architecture and learned representations could be adapted for other critical clinical tasks, such as classification (e.g., mortality prediction on the CinC 2012 dataset) or patient phenotyping? My confidence in the paper's central claim would increase significantly if you could provide even a preliminary result showing that MIRA's embeddings can be effectively used for a non-forecasting task.

3. The pre-training corpus is heavily weighted towards the MIMIC-III dataset (88.06% of time points), which originates from a single US medical center. This raises questions about potential institutional or demographic biases in the pre-trained model. To what extent do you believe this affects the model's zero-shot generalizability? To better demonstrate the value of pre-training, could you conduct an experiment where MIRA is fine-tuned on a small fraction of an out-of-distribution dataset and compare its sample efficiency against a strong baseline trained from scratch on that same small dataset? A result showing that pre-trained MIRA learns faster or better with less data would provide powerful evidence for its utility as a transferable foundation for diverse clinical settings.

**Ethical Concerns:**

["NO or VERY MINOR ethics concerns only"]

**Final Justification:**

Authors thoroughly answered all my questions.

**Limitations:**

Yes

**Quality:**

3

**Strengths And Weaknesses:**

Strengths

- The paper addresses the highly relevant but underserved problem of creating a foundation model specifically for medical time series data. The proposed MIRA architecture is original and well-motivated, combining several components to specifically target the core challenges of irregular sampling, missing values, and heterogeneous data common in healthcare.

- The quality of the empirical evaluation is a major strength. MIRA is tested against a wide range of state-of-the-art zero-shot and full-shot models on diverse clinical benchmarks. The results are strong and consistently demonstrate superior performance, which is further validated by thorough ablation studies and robustness checks against data missingness.

- In addition to the model itself, the work provides two significant resources for the research community: a large, curated pre-training corpus of over 454 billion medical time points and a new, comprehensive benchmark for clinical forecasting tasks.

- The paper is well-written, clearly structured, and easy to follow. The authors do an excellent job of motivating the problem and presenting their complex architecture in an understandable manner, aided by effective diagrams.

Weaknesses

- The paper frames MIRA as a "unified foundation model," but the evaluation is exclusively focused on forecasting tasks. While the motivation mentions other clinical tasks like phenotyping and classification, the experiments do not test MIRA's capabilities on these fronts (e.g., via fine-tuning). Demonstrating its utility beyond forecasting would be necessary to fully substantiate the "unified foundation model" claim.

-  The model architecture is inherently complex, integrating a transformer, MoE layers, and a Neural ODE solver. The use of a numerical ODE solver, in particular, can be computationally expensive. The paper lacks a detailed analysis of MIRA's inference speed, memory footprint, and training cost compared to the baselines. This information is crucial for understanding its practicality for real-time clinical applications.

- The pre-training corpus is built from publicly available datasets, which is excellent for reproducibility. However, the paper could provide a more in-depth discussion of the potential demographic or institutional biases within these datasets (e.g., MIMIC-III is from a single medical center) and how such biases might impact the model's performance and fairness when deployed in different clinical environments.

- The main results in Table 2 compare zero-shot MIRA against fine-tuned specialist models. While MIRA's ability to compete with or even exceed these models in a zero-shot setting is impressive, the comparison is not entirely direct. Fine-tuning MIRA on these downstream tasks would provide a more complete picture, helping to disentangle the benefits of its pre-training from its architectural advantages in a full-data scenario.

---

> ### Author Rebuttal · Authors · 2025-07-31
>
> We sincerely thank the reviewer for their thoughtful and constructive feedback on our submission. We are especially appreciative of your recognition of the motivation and technical contributions of MIRA in addressing the challenges of medical time series. In response to your suggestions, we have made the following updates:
>
> 1. **Quantified MIRA’s computational efficiency.**
> 2. **Explore the potential for adapting MIRA to classification tasks**, demonstrating its broader applicability beyond forecasting.
> 3. **Investigated potential dataset imbalance and bias**, using influence score analysis and analysis of model menchinism.
> 4. **Demonstrated MIRA’s sample-efficient domain adaptation**, by comparing few-shot fine-tuning with from-scratch training on an out-of-distribution dataset.
>
> We hope these additions clarify the contributions and practical value of our work. Please see below for our point-by-point responses.
>
> ---
>
> **Q1: Could you provide a detailed analysis of MIRA’s computational footprint, including inference latency (seconds per forecast window) and memory requirements, directly compared to TimesFM and ContiFormer?**
>
> A: We measured average inference latency and GPU memory consumption over 10 batches (batch size = 32) with history length 512 and prediction length 96:
>
> |  | MIRA | TimesFM | Contiformer |
> | --- | --- | --- | --- |
> | Inference Speed | 0.5119s | 1.0702s | 1.0351s |
> | Peak GPU memory | 2575.65MB | 4489.0 MB | 1072.11 MB |
>
> MIRA achieves 2× faster inference than TimesFM (500m) despite comparable model complexity, demonstrating superior parameter efficiency through sparse MoE activation. Although ContiFormer's memory usage is smaller, it requires iterative extrapolation for multi-step forecasting, significantly increasing inference time compared to the other two models.
>
> **Q2: How can MIRA’s architecture or learned embeddings be adapted to non-forecasting clinical tasks (e.g., mortality prediction on the CinC 2012 dataset or patient phenotyping), and can you share any preliminary downstream results?**
>
> A: We use HeartBeat (contains ECG-based labels for normal and abnormal rhythms) from CinC 2016 as classification dataset to evaluate whether our proposed model could be adopt to other task format. We keep the same data split and metrics reported in Time Series Library (TSLib) and baselines[1,2]. We extract the final hidden states from MIRA as sequence-level representations and pass them to a lightweight classifier (e.g., CNN variant) for training.
>
> Our results show that MIRA achieves competitive accuracy compared to specialized models. These findings suggest that MIRA’s learned embeddings retain clinically useful information and can be effectively adapted for non-forecasting tasks. We will extend this line of work to additional downstream tasks and datasets.
>
> |  | TimesNet | DLinear | Autoformer | MIRA |
> | --- | --- | --- | --- | --- |
> | ACC | 78.0 | 75.1 | 74.6 | 77.6 |
>
> **Q3.1: Could MIMIC-III’s 88% share in the pre-training corpus introduce demographic or institutional biases into MIRA’s representations?**
>
> A:  Thank you for this insightful question. While MIMIC-III play a major role in pre-training corpus, we believe the risk of demographic or institutional bias in MIRA is minimal, for the following reasons:
>
> First, MIRA operates solely on physiological time series (e.g., vitals, labs, ECG) and is not provided with age, sex or hospital information. This prevents the model from conditioning on potential sources of bias and encourages learning representations based only on universal temporal patterns present in the data.
>
> Second, MIRA aim to autonomously learn latent temporal patterns from physiological signals, rather than memorizing dataset-specific artifacts. By design, the model encourages generalization across diverse domains, and the MoE architecture facilitates this by dynamically routing inputs based on their signal properties, not dataset identity. This enables MIRA to develop domain-robust representations that transfer well beyond any single data source.
>
> Our empirical results support this: For example, table 2 shows that the performance is still comparable to other datasets. If institutional bias were significant, we would expect substantial performance degradation on these datasets, which we do not observe. Additionally, please also refer to reviewer vkqd’s Q1.1, where we conduct the influence score of training dataset to quantify their impact. Result shows that although MIMIC-III constitutes a large portion of the pretraining corpus, it does not overwhelmingly influence the learned representations, suggesting that potential biases introduced by MIMIC-III are limited.
>
> **Q3.2:** **Can you provide few-shot fine-tuning results on an out-of-distribution dataset to compare MIRA’s sample efficiency against a model trained from scratch?**
>
> A: We constructed a training set with 20k samples based on MIT-BIH dataset, each having a sequence length of 256. During inference, the prediction lengths are set to 24, 36, 48, and 64, with the context length being twice the prediction length. Average RMSE performance reported.
>
> Notably, **Time-MoE required 80% of the data (~16k samples)** to reach similar RMSE levels as MIRA with 128–256 samples. This underscores MIRA’s strong few-shot performance, as well as its potential to **significantly reduce data and computational costs** in new clinical settings.
>
> | Few-shot | 16 | 32 | 64 | 128 | 256 |
> | --- | --- | --- | --- | --- | --- |
> | MIRA | 0.1224 | 0.1221 | 0.1220 | 0.1215 | 0.1211 |
> | Training Size | 20% | 50% | 60% | 70% | 80% |
> | Time-MoE | 0.2004 | 0.1559 | 0.1368 | 0.1222 | 0.1206 |
>
> [1] Wu, Haixu, et al. "Timesnet: Temporal 2d-variation modeling for general time series analysis." arXiv preprint arXiv:2210.02186 (2022).
>
> [2] Wang, Yuxuan, et al. "Deep time series models: A comprehensive survey and benchmark." arXiv preprint arXiv:2407.13278 (2024).
>
> [3] Haq, Kazi T., et al. "Demographic and methodological heterogeneity in electrocardiogram signals from guinea pigs." *Frontiers in physiology* 13 (2022): 925042.

---

> > ### Comment · Reviewer_LN9n · 2025-08-09
> > **Response to rebuttal**
> >
> > I want to thank the authors for thoroughly answering my questions, I am quite happy with the new results and discussion.

---

> ### Comment · Area_Chair_RRz3 · 2025-08-09
> **Urgently provide feedback**
>
> Dear LN9n,
> it is mandatory you provide your feedback

---

### Note · Authors · 2025-08-13

Dear Area Chair and Reviewers,

We provide this summary to assist with your meta-review process by outlining our main contributions, the key points addressed in rebuttal, and the outcomes of our reviewer discussions.

**Summary of our contribution**
We present MIRA, a unified foundation model for irregular medical time series forecasting. MIRA integrates Continuous-Time Rotary Positional Encoding (CT-ROPE), a frequency-specialized MoE layer, and a Neural ODE-based extrapolation block, enabling accurate forecasting at arbitrary timestamps across diverse clinical domains. Pre-trained on a large medical corpus, it achieves strong zero-shot results on in- and out-of-distribution benchmarks.

**Outcomes of our discussions with reviewers**
All three reviewers recognized the novelty of our methods. Concerns raised during the review, including computational efficiency, applicability beyond forecasting, dataset imbalance and bias, robustness to block-missingness, statistical significance testing, and interpretability, were directly addressed through new experiments, analyses, and detailed explanations.

• **Reviewer LN9n**: *Raised points on computational efficiency, model generality, and bias; was satisfied after we provided computational benchmarks, classification results, few-shot adaptation experiments, and an influence score analysis.*

• **Reviewer vk9d**: *Expressed concerns about pre-training dataset imbalance, heterogeneous sampling, and robustness; was satisfied after we provided an influence score analysis, a block-missingness evaluation, and statistical tests.*

• **Reviewer CKRd**: *Requested deeper discussion, qualitative examples, and computational efficiency analysis; was satisfied after we provided expanded ablation studies, a qualitative case study, and an efficiency discussion.*

**Summary of rebuttal**

All reviewers explicitly confirmed that their concerns were resolved and maintained support for acceptance. We are grateful for the constructive feedback, which has strengthened the clarity and completeness of this work, and believe that the constructive reviewer dialogue and substantial additional evidence provided during rebuttal have further improved its quality.

---

### Decision · Program_Chairs · 2025-09-17

**Decision:**

Accept (poster)

**Comment:**

The paper presents MIRA, a novel foundation model tailored for medical time series data, and it stands out for its originality and relevance. The authors tackle a pressing challenge in healthcare AI: handling irregular, heterogeneous, and incomplete time series data. MIRA’s architecture is thoughtfully designed, integrating components like CT-RoPE, temporal Mixture-of-Experts (MoE), and Neural ODEs to address these issues. The empirical evaluation is particularly strong, with MIRA outperforming several state-of-the-art models across diverse benchmarks. The inclusion of a massive pre-training corpus and a new benchmark suite further enhances the paper’s contribution to the field. Moreover, the clarity of writing and the availability of code and datasets support reproducibility and accessibility for future research. The paper in its first submission suffered several limitations. While MIRA is positioned as a “unified foundation model,” its evaluation is narrowly focused on forecasting tasks. This leaves its broader applicability, such as classification or phenotyping, unexplored, weakening the claim of generality. The architecture’s complexity, especially the use of Neural ODEs, raises concerns about computational efficiency, yet the paper lacks a detailed analysis of inference speed, memory usage, and training cost. This omission is critical given the real-time demands of clinical settings. Additionally, although the pre-training data is publicly available, the paper does not sufficiently address potential biases stemming from sources like MIMIC-III, which could affect fairness and generalizability. Another notable gap is the lack of interpretability tools, essential for clinical adoption, since clinicians need to understand and trust model outputs. Finally, the discussion section is underdeveloped; it misses an opportunity to critically analyze why MIRA’s components lead to performance gains, which would have added depth to the paper’s insights.
However, the rebuttals were very clear and answered point by point to criticisms from the reviewers, that all recognized their concerns were all fully solved after the rebuttal from the authors and the subsequent highly engaging discussion. All reviewers confirmed they were fully satisfied with the discussion.